# Slower growth of *Escherichia coli* leads to longer survival in carbon starvation due to a decrease in the maintenance rate

Elena Biselli[1,†] iD, Severin Josef Schink[1,2,†] iD & Ulrich Gerland[1,*] iD

## Abstract

Fitness of bacteria is determined both by how fast cells grow when nutrients are abundant and by how well they survive when conditions worsen. Here, we study how prior growth conditions affect the death rate of *Escherichia coli* during carbon starvation. We control the growth rate prior to starvation either via the carbon source or via a carbon-limited chemostat. We find a consistent dependence where death rate depends on the prior growth conditions only via the growth rate, with slower growth leading to exponentially slower death. Breaking down the observed death rate into two factors, maintenance rate and recycling yield, reveals that slower growing cells display a decreased maintenance rate per cell volume during starvation, thereby decreasing their death rate. In contrast, the ability to scavenge nutrients from carcasses of dead cells (recycling yield) remains constant. Our results suggest a physiological trade-off between rapid proliferation and long survival. We explore the implications of this trade-off within a mathematical model, which can rationalize the observation that bacteria outside of lab environments are not optimized for fast growth.

**Keywords** bacterial fitness; bacterial survival; bacterial systems biology; death rate; quantitative physiology

**Subject Categories** Metabolism; Microbiology, Virology & Host Pathogen Interaction

**Mol Syst Biol. (2020) 16: e9478**

## Introduction

Bacteria are exposed to a variety of environments, from stressful and nutrient-poor to ideal and nutrient-abundant. The average proliferation of bacteria through cycles of famine and feast, i.e., their fitness, depends not only on the ability to grow rapidly, but also on the ability to survive when conditions worsen. While some bacteria can produce dormant endospores that can survive for thousands of years (Vreeland *et al*, 2000; Setlow, 2007), vegetative bacteria cannot. These bacteria require continuous maintenance to sustain basic cellular functions and prevent cell death. It is clear that the ability to reduce and optimize the maintenance rate is crucial for maximizing survival in limited environments. But how cells could achieve this feat is still largely unclear.

Measurements of the maintenance rate are usually performed during exponential growth (Pirt, 1965), when nutrients are still abundant. Such measurements overestimate the basic maintenance of non-growing cells (Hoehler & Jørgensen, 2013) and hence do not characterize the state of starved cells. The maintenance rate during carbon starvation was recently measured for *Escherichia coli* and shown to be a key determinant of the death rate of the cell population (Schink *et al*, 2019). Which factors determine the maintenance rate and to which extent a bacterial species can adapt its maintenance rate to increase its fitness during starvation are still unclear (Hoehler & Jørgensen, 2013). Here, we focus on these questions, using *E. coli* as a model organism.

A key signal for upcoming starvation is a decrease in growth rate: When a cell culture approaches starvation, nutrient concentrations fall below the substrate affinity, such that nutrient uptake decreases and growth slows down (Monod, 1949). Many physiological properties of a cell depend on growth rate, e.g., cell size and ribosome content (Schaechter *et al*, 1958), but also the detailed composition of the proteome (Scott *et al*, 2010; Hui *et al*, 2015; Schmidt *et al*, 2016). Therefore, one may expect that the ability of cells to survive also depends on the growth conditions prior to starvation. A priori, it is not clear whether slow growth or fast growth prior to starvation would be beneficial for survival. In slow growth, the general stress response is upregulated (Kolter & Siegele, 1993; Hengge, 2011), which could help *E. coli* to survive. On the other hand, in fast growth, more nutrients are available, e.g., for glycogen storage (Fung *et al*, 2013).

In order to study how the growth conditions affect the subsequent survival of cells, we vary the bacterial growth rate in two different ways: either by culturing *E. coli* in media with different carbon substrates, where each substrate supports a certain growth rate, or by varying the dilution rate in a carbon-limited chemostat.

1 Physics of Complex Biosystems, Physics Department, Technical University of Munich, Garching, Germany
2 Department of Systems Biology, Harvard Medical School, Boston, MA, USA
*Corresponding author. Tel: +49-89-28912394; E-mail: gerland@tum.de
†These authors contributed equally to this work

We then rapidly deprive each culture of its carbon substrate (i.e., its energy source) and measure its survival kinetics, i.e., the remaining density of viable cells at different time points. This approach allows us to measure quantitative changes in death rate as a function of the steady-state growth conditions, as opposed to the classical "entry into stationary phase" of the bacterial life cycle, where bacteria continuously adapt as the nutrient quality of the medium worsens (Monod, 1949; Hengge, 2011).

During starvation, the resource which bacteria use for maintenance is nutrients recycled from dead bacteria. As a result, both changes in maintenance rate or recycling yield can alter the survival kinetics. To dissect the individual contributions of maintenance rate and recycling yield to death rate, we use the quantitative approach of Schink *et al* (2019).

## Results

### Death rate of starved culture depends exponentially on rate of growth prior to starvation

We grow *E. coli* in media with different carbon substrates (Fig 1A), obtaining exponential growth at a rate (slope of the lines in Fig 1A) that depends on the carbon substrate. Between rich medium and minimal medium supplemented with a poor carbon substrate, the data display a 15-fold change in growth rate (see Table EV1 for all values). Once the cultures have reached an optical density ($OD_{600}$) of about 0.5, we wash and resuspend the cells in carbon-free medium (see Methods). This step removes left-over nutrients from complex media like LB or fermentative by-products such as acetate and ensures that bacteria are starved of all external carbon substrates. We then follow the survival kinetics by measuring bacterial viability via plate counting at different time points after carbon starvation (Fig 1B). Bacterial viability measurements by live/dead staining yield survival kinetics comparable to plate counting (Schink *et al*, 2019). The number of "colony-forming units" (CFU) per ml decreases exponentially for all cultures. The death rate (slope of the lines in Fig 1B) depends on the carbon substrate used for growth, with cultures grown on LB dying more than 5 times faster than those on mannose (see Table EV1 for all values).

Figure 1C plots the death rates $\gamma$ of the five cultures in Fig 1B and of other cultures against their growth rates $\mu$, revealing a clear trend of slower growing cultures dying more slowly than faster growing cultures. The quantitative dependence, $\gamma(\mu)$, is well described by an exponential fit, $\gamma = (0.23 \pm 0.01)$ h$^{-1}$ exp ($\mu$ $(0.87 \pm 0.13)$ h), with a goodness-of-fit parameter $Q = 1.0$ (probability that a chi-square value of this magnitude did not arise by chance, see Methods for details of fitting procedure). This dependence is remarkable for two reasons: First, the death rate appears not to depend in a specific way on the carbon used for growth, but instead only on the growth rate supported by the carbon substrate. This is consistent with Schaechter, Maaløe, and Kjeldgaard's seminal finding that the cellular composition does not depend on the specifics of the nutrient composition, but rather on the resulting growth rate (Schaechter *et al*, 1958). Second, other bacterial properties also scale exponentially with growth rate. Cell volume during growth, in particular, increases exponentially with growth rate (Schaechter *et al*, 1958).

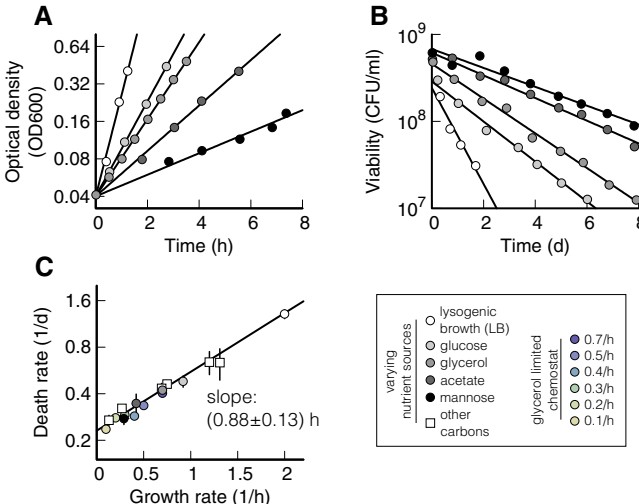

**Figure 1. Growth–death dependence.**

A Exponential growth of *Escherichia coli* K-12, measured using optical density ($OD_{600}$), in lysogenic broth (LB, white circles) or minimal medium supplemented with different carbon sources, listed in the legend on the bottom right in a grayscale. All the cultures are grown in batch mode, see Methods. Growth rates (slope of the exponential fits) are listed in Table EV1.

B Bacterial viability in colony-forming units (CFU) per ml of cultures from panel (A), grown until $OD_{600}$ 0.5, washed, and re-suspended in carbon-free minimal medium. Death rates (slopes of the exponential fits) are listed in Table EV1. Initial density for fast growth (lighter shades) is lower than for slow growth (darker shades), due to larger cell sizes that lead to lower cell densities per $OD_{600}$. Note that after the exponential decay shown here, mutants gradually take over and long-term survival can last months to years (Steinhaus and Birkeland, 1939, Zambrano et al, 1993, Finkel, 2006).

C Death rates of panel (B) plotted against growth rates of panel (A) shown as circles in a grayscale. Squares show data obtained using eight other carbon substrates, also listed in Table EV1. Color circles show data obtained by varying growth rate in a chemostat. The black line shows a linear weighted least square fit to log-transformed data that takes uncertainty in data points into account (see Methods). Data shown as mean ± (standard deviation) SD. Two or three replicates per condition.

Next, we aim to narrow down the origin of the death rate dependence on growth rate. It was previously established that death rate in batch culture is quantitatively determined by the ratio of maintenance rate to recycling yield (Schink *et al*, 2019). Figure 2 explicitly shows this relation (Fig 2A) and illustrates the assays for both quantities (Fig 2C and G). To investigate whether the death rate dependence in Fig 1C is due to a change in the maintenance rate, the recycling yield, or both, we perform a set of experiments where we measure death rate, recycling yield, and maintenance rate in varying growth conditions.

### Varying growth rate on a single carbon substrate in a chemostat

Maintenance rate is measured by quantifying how long cell death halts after a small concentration of a carbon source is supplied during starvation (Fig 2G). In order for maintenance measurements of different cultures to be quantitatively comparable, this carbon source needs to be the same for all experiments. For this reason, we turn to a chemostat setup to change growth rate on the same carbon

substrate. We choose glycerol as the limiting nutrient in the medium and let the bacterial culture grow in steady state at fixed dilution rate, such that dilution rate equals growth rate (see Methods). In this setup, glycerol concentration will be constant and low enough to reduce growth rate (< 5 μM), but bacterial density can be kept comparable to batch experiments ($5 \cdot 10^8$ CFU ml$^{-1}$). We use the chemostat to vary growth rate from 0.1 to 0.7 h$^{-1}$. For faster growth, we use a GlpK22 (NQ898) mutant without catabolic repression of the GlpK enzyme (Pettigrew *et al*, 1996) that grows 30%

faster than wild type in batch culture in minimal medium supplemented with glycerol ($\mu_{\text{GlpK22}} = 0.89$ h$^{-1}$). As a control, wild-type cells (WT) are grown in batch culture in glycerol minimal medium ($\mu_{\text{WT}} = 0.70$ h$^{-1}$). After at least six generations in steady-state growth, samples of cells in the chemostat are extracted, washed, and starved in carbon-free minimal medium (see Methods). GlpK22 and wild type are grown in batch cultures, and then also washed and starved in carbon-free medium. During starvation, we observe that viability decreases exponentially, with death rates ranging from

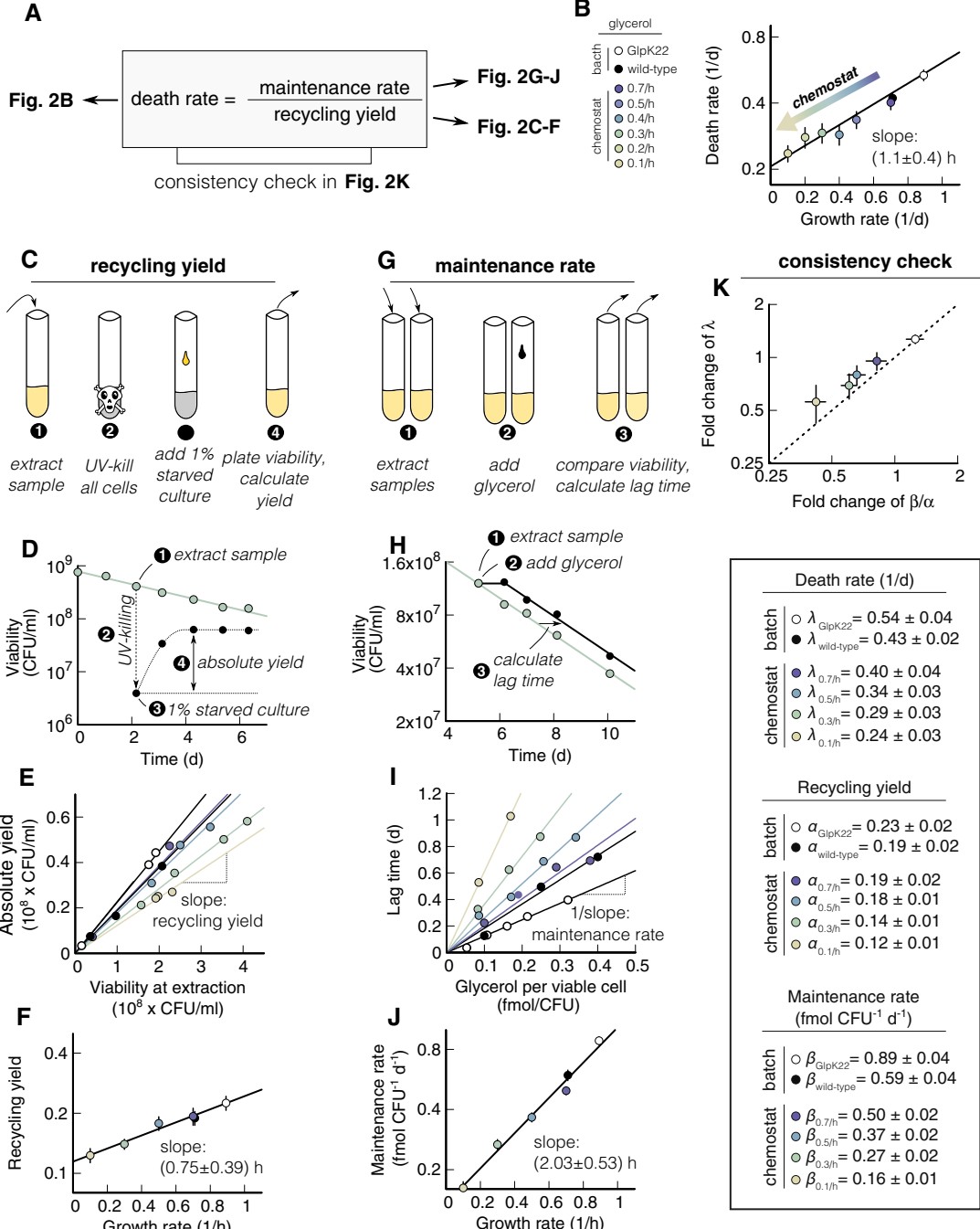

**Figure 2.**

**Figure 2. Quantification of maintenance rate and recycling yield of *Escherichia coli*.**

- A   In carbon starvation, death rate, $\gamma$, is determined by ratio of maintenance rate $\beta$ to the recycling yield $\alpha$ (Schink *et al*, 2019).
- B   Death rates of wild-type and GlpK22 mutants plotted versus growth rates. Wild type is grown in glycerol minimal medium in batch cultures (black circle) and in "chemostat" continuous cultures with growth rates coded in colors, see legend on the center right. GlpK22 mutants (white circle) are grown in glycerol minimal medium in batch culture. Data shown as mean $\pm$ (standard deviation) SD. Two replicates per condition. The black line shows the fit of Fig 1C.
- C   Graphical synopsis of the assay to measure recycling yield: At different times during starvation, a sample is (1) extracted from a starved culture, (2) sterilized with UV-light, (3) inoculated with 1% of the original starved culture, and (4) growth is measured using plate counting.
- D   Example experiment of the assay sketched in panel (C). Cells previously grown at 0.3 $h^{-1}$ in continuous culture are starved (green circles). After 2 days of starvation, a sample is (1) extracted and (2) UV-sterilized, see dashed line, followed by (3) inoculation of 1% of starved and (4) growth of the culture. The difference between the maximum viability after growth and the initial viability at point (3) is the absolute yield.
- E   The absolute yield measured at three different time points for each of the growth conditions shown in panel (B) is plotted against the viability of the starvation culture at the extraction time. The recycling yield is extracted as the slope of the linear fits.
- F   Recycling yield from panel (E) plotted versus growth rate. Data shown as mean $\pm$ SD. Two replicates per condition.
- G   Graphical synopsis of the assay to measure the maintenance rate. At one point during starvation, a sample (1) extracted from a starvation culture, and split into several tubes. (2) Different concentrations of glycerol are added to each tube, small enough to support survival, but not growth and (3) viability are measured.
- H   Example experiment from a starved culture previously grown at a rate of 0.3 $h^{-1}$ with a viability of 1.21·$10^8$ CFU $ml^{-1}$ after 5 days of starvation. After (1) extraction and (2) addition of 40 μM of glycerol, the decay of viability is delayed (black circles) compared to a control without glycerol (green symbols). Per viable cell, the glycerol addition in this experiment is 0.33 fmol $CFU^{-1}$. After an initial period of survival, the culture with added glycerol (black) dies at the same rate as the control (green). The "lag time" fitted to these data is 1 day.
- I   Lag time of the experiments of panel (B) for different glycerol concentrations. Lag increases linearly with glycerol concentration, and maintenance rate is extracted as the inverse of the slope of the linear fits.
- J   Maintenance rate values extracted from panel (I) plotted versus growth rate. Data shown as mean $\pm$ SD. Two replicates per condition.
- K   Consistency check of the equation in panel (A). Fold changes in maintenance rate, FC($\beta$), divided by the fold changes in the recycling yield, FC($\alpha$), plotted versus the fold changes in death rate, FC($\gamma$). All fold changes are relative to wild type in batch, growing at 0.7 $h^{-1}$. The dashed line shows the unity line FC($\gamma$) = FC($\beta$)/FC($\alpha$).

Data information: Symbol color is identical in all panels and depicted in the legend on the right. Table EV2 shows growth rate, yield, and maintenance rate for all experiments. Solid black lines show linear fits (on log transformed data, if necessary), using weighted least square fits. Slopes are reported with standard error and take uncertainty in data into account when necessary (see Methods).

0.25 to 0.59 $day^{-1}$ (Fig EV1 and Table EV1). The dependence of the death rate on the growth rate (Fig 2B) is again well described by an exponential fit (black line, $\gamma = (0.21 \pm 0.02)$ $h^{-1}$ exp ($\mu$ (1.1 $\pm$ 0.4) h), $Q = 0.99$). These findings confirm that the growth–death correlation is independent of specific medium composition, but instead dependent only on growth rate.

**Change in recycling yield with pre-starvation growth rate**

Next, we determined the recycling yield, which is defined as the fraction of nutrients that a viable cell can scavenge from a dead cell. The recycling yield can be quantified by measuring to what density bacteria grow when dead bacteria are provided as nutrients (Schink *et al*, 2019). The assay is sketched in Fig 2C and described in Methods. Briefly, at different times during starvation, we (i) extract a sample of a starved culture, (ii) UV-kill all cells in the sample, (iii) add 1% of the original starved culture with viable cells, and (iv) measure regrowth by plating. Figure 2D shows one exemplary measurement for bacteria previously grown at 0.3 $h^{-1}$ in the chemostat (green symbols). The difference between the maximal regrowth and the initial viability in the sample is the "absolute yield", see symbol (iv) in Fig 2D. In Fig 2E, we show three measurements of the absolute yield plotted against the viability at extraction six of the growth experiments from Fig 2B. In this plot, the recycling yield is the slope of the linear fit, i.e., the ratio of absolute yield and viability at extraction. It ranges from 12% for the slowest growth to 23% for the fastest growth (see Table EV2). This value represents the number of cells that can grow on one dead cell, i.e., a recycling yield of 12% means that about eight dead cells are needed to produce one new cell. The recycling yield displays a dependence on growth rate that is well described by an exponential behavior, $\alpha = (0.11 \pm 0.01)$ exp ($\mu$ (0.78 $\pm$ 0.35)h), $Q = 0.997$ (Fig 2E).

**Change in maintenance rate with pre-starvation growth rate**

The maintenance rate is the second determining factor of the death rate (Fig 2A). The lower the maintenance rate, the fewer nutrients per unit time a cell has to consume to remain viable. The maintenance rate can be quantified by adding a small amount of nutrient to a starved culture and measuring the resulting time delay in the decay of cell viability (Schink *et al*, 2019). The assay is sketched in Fig 2G and described in Methods. Briefly, at one point during starvation, we (i) extract several samples, (ii) add different concentrations of glycerol, and (iii) compare viability between samples. Figure 2H shows an exemplary experiment for cells previously grown at a rate of 0.3 $h^{-1}$ in the chemostat, where the addition of 40 μM glycerol allows a culture (black symbols) to survive more than 1 day longer than the control culture (green symbols). The delay in the survival curves between the two experiments is the "lag time", see symbol (3) in Fig 2H. In Fig 2I, we show the lag time for different growth experiments and different glycerol concentrations. In this plot, the maintenance rate is the inverse of the slope of the linear fits. It ranges from 0.16 fmol $day^{-1}$ $CFU^{-1}$ for the slowest growth to 0.81 fmol $day^{-1}$ $CFU^{-1}$ for the fastest growth. The value of the maintenance rate represents the number of glycerol molecules a single cell needs to survive 1 day. Just as volume and recycling yield, also the maintenance rate $\beta$ displays an apparent exponential dependence on growth rate, see Fig 2J, but with a slope of about twice that of either ($\beta = (0.14 \pm 0.02)$ exp ($\mu$ (1.88 $\pm$ 0.56)h), $Q = 0.99$).

We can compare our measurement of maintenance rate to literature values. Using a conversion of 15 ATP per molecule glycerol (Kaleta *et al*, 2013), a dry mass per OD of 509 μg $ml^{-1}$ $OD_{600}^{-1}$ (Erickson *et al*, 2017), and $10^9$ CFU $ml^{-1}$ $OD_{600}^{-1}$, we calculate that a maintenance rate of 0.5 fmmol (glycerol) $day^{-1}$ $CFU^{-1}$ corresponds to 0.61 mmole of ATP per ((g dry weight) h). In comparison,

estimates for non-growth-associated maintenance energy during exponential growth for *E. coli* are about 10-fold higher at 7.6 (Varma & Palsson 1994) and 8.4 mmole of ATP per ((g dry weight) h) (Feist *et al* 2007). This shows that maintenance rate measurements during nutrient abundance are not readily convertible to starvation.

## Changes in maintenance rate and recycling yield explain change in death rate

Next, we test if the changes in maintenance rate and recycling yield are consistent with the changes observed in the death rates. According to the relation in Fig 2A, death rate $\gamma$ is set by maintenance rate $\beta$ divided by recycling yield $\alpha$. In Fig 2K, we thus plot the fold change in the death rate, FC($\gamma$), against the fold change in maintenance rate, FC($\beta$), divided by the fold change in the recycling yield, FC($\alpha$). Each change is relative to the death rate, maintenance rate, and recycling yield of the control culture: wild type previously grown in batch with glycerol (black symbols throughout Fig 2). The dotted line indicates the unity line, FC($\gamma$) = FC($\beta$)/FC($\alpha$), corresponding to the prediction by the relation in Fig 2A. The data are well described by the predicted line, indicating that our measured maintenance rates and recycling yields indeed capture the change in death rate.

## Change in cell volume with pre-starvation growth rate

One challenge with interpreting the measured recycling yields and maintenance rates is that both parameters can depend on cell size. The bigger a dead cell is, the more recyclable nutrient it could contain. The bigger a viable cell is, the more nutrients it might consume for maintenance. Since the average cell size during starvation is generally smaller than during growth, we perform cell size measurements during starvation conditions (Fig 3). We determine length and width of individual starved bacteria from phase-contrast images (see Methods). The average length and width increase exponentially with growth rate, with slopes of (0.23 ± 0.18) h and (0.32 ± 0.21) h, respectively (Fig 3A and Table EV3). From the individual cell lengths and width, we calculate the volume of each cell by assuming cells have a perfect rod shape (top of Fig 3B). The average cell volume then increases exponentially with growth rate according to $V = (0.43 \pm 0.04)\ \exp\ (\mu\ (0.88 \pm 0.33)\text{h})$, $Q = 0.99$, see Fig 3B and Table EV3.

It is noteworthy that the logarithmic slopes of the volume dependence on growth rate (Fig 3B) and the recycling yield dependence on growth rate (Fig 2F) are indistinguishable within our experimental uncertainty. Consequently, normalizing recycling yield to cell volume results in a constant normalized yield, independent of growth rate within the statistical error (Fig 3C). Maintenance rate, on the other hand, still increases significantly when normalized to cell volume (Fig. 3D), with a logarithmic slope of (1.0 ± 0.5) h. This means that while bigger cells contain proportionally more nutrients, they use more than one would expect from their size for maintenance. This implies that there are factors involved beyond cell size that modulate the maintenance requirement of *E. coli*.

## Growth–death dependence does not hinge on RpoS

A sigma factor that is often implicated in the regulation of the maintenance rate of *E. coli* is the general stress response regulator RpoS

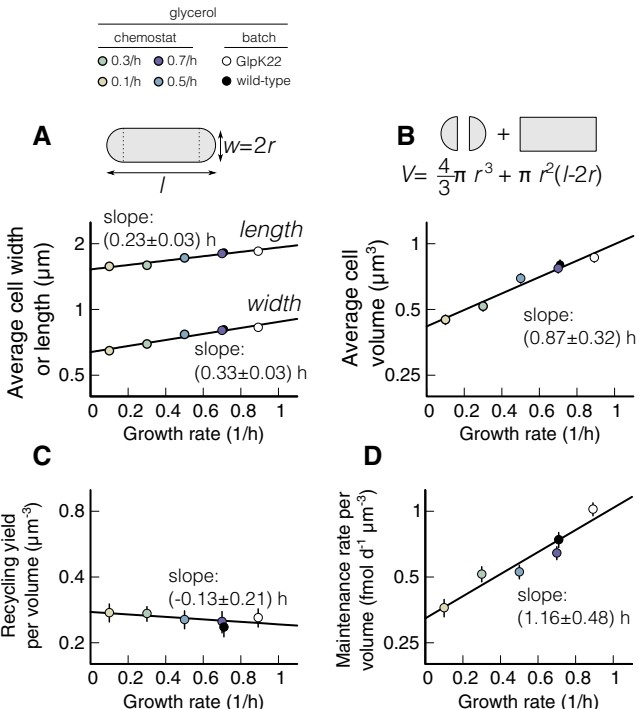

**Figure 3.   Normalization of maintenance rate and recycling yield with cell volume of *Escherichia coli*.**

A, B   Cell size during starvation. Length and width of the cells are measured with phase-contrast microscopy, and the volume is computed considering cell shape as a cylinder with two semi-spheres, as described in the graphical synopsis at the top (see also Table EV3 and Methods). Each measurement is an average of 200 cells. (A) Measured length and width of wild-type cells starved in batch cultures and previously grown in the chemostat at different steady-state growth rates and of GlpK22 cells starved and previously grown in batch culture at $\mu = 0.9\ h^{-1}$ (see upper color legend). Note that, as shown in Schink *et al* (2019), cell widths do not change from steady-state growth to starvation, while lengths decrease. Both length and width increase exponentially with growth rate. (B) Starvation volume of the cells described in panel (A), computed as explained in the graphical synopsis (see also Methods). In agreement with literature (Schaechter *et al*, 1958), it increases exponentially with growth rate (black line).

C, D   Recycling yield and maintenance rate per cell volume. (C) Recycling yields and (D) maintenance rate measured as described in Fig 2 are normalized per cell volume and plotted versus the previous growth rate of the cultures they refer to (see color legend at the top and Table EV3). The normalized yield is constant within the uncertainty. Maintenance rate increases significantly with a slope of (0.99 ± 0.55) h, matching the increase in the death rate, (1.1 ± 0.4) h, shown in Fig 2B. Solid black lines show linear fits on log-transformed data, using weighted least square fits.

Data information: Slopes are reported with standard error and take uncertainty in data into account (see Methods). Data shown as mean ± SD. Two replicates per condition.

(Hengge-Aronis, 1993; Hengge, 2011; Franchini *et al*, 2015). RpoS is up-regulated during slow growth, and the expression of almost 500 genes is correlated with its abundance (Hengge, 2011). To test whether RpoS plays a key role in the growth–death dependence, we grow *rpoS* gene knock-out mutants ($\Delta rpoS$) in batch cultures using the same carbon substrates we used in Fig. 1C. If *rpoS* was responsible for the growth–death relation, its knock-out should abolish this

relationship. Figure 4 plots death rates against growth rates for *ΔrpoS* (gray symbols), together with the wild-type data from Fig. 1 for reference (white symbols). The individual death rates for *ΔrpoS* are shown in Table EV4. We find that the correlation between death rate and growth rate persists for *ΔrpoS*, but the logarithmic slope decreases (*ΔrpoS*: $0.57 \pm 0.13$ h, $Q = 0.96$, compared to WT: $0.87 \pm 0.13$ h). Because death rate still increases with growth rate, we conclude that RpoS cannot be solely responsible for modulating maintenance rate during starvation.

## Mathematical model of a proteome-based growth–death coupling

The observation that the RpoS regulon, which by itself includes dozens of genes, accounts for only a fraction of the observed maintenance rate variation suggests that it is only part of a major proteome remodeling that affects the maintenance rate of cells. Proteome remodeling, with the abundance of major portions of the cellular proteome modulated depending on the growth state, is well characterized in *E. coli*. For instance, ribosomal and ribosome-affiliated proteins can take up somewhere between 10 and 40% of the proteome, depending on growth rate (Scott *et al*, 2010). As this "R-sector" of the proteome increases, other parts of the proteome decrease in relative abundance (Scott *et al*, 2010; Hui *et al*, 2015). This coupling of proteome fractions implies a trade-off, whereby the proteome composition can either favor fast growth or long survival. The signature of this trade-off is the growth–death correlation of Fig 1. To understand the evolutionary implications of this trade-off, we turn to a mathematical model for a proteome-based growth–death coupling.

We consider a minimal ecological scenario of periodic phases of feast and famine, as depicted in Fig 5A. In this scenario, bacteria grow for a limited time $T_+$, followed by starvation for a time $T_-$. We regard the durations of growth and starvation as ecological parameters and ask how bacterial fitness depends on these parameters as well as their growth rate. We assume that bacteria cannot independently optimize growth and death rate, but are instead forced to trade-off growth and death rates according to the exponential relation of Figs 1 and 2, $\gamma(\mu) = 0.21$ day$^{-1}$ exp (1.0 h $\mu$). In this case, they will first grow exponentially during feast, starting from

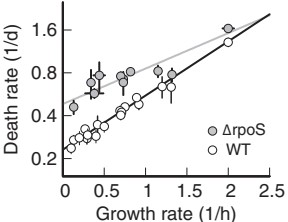

**Figure 4. Influence of RpoS knock-out on the death rate dependence.**

Death rates of *ΔrpoS* mutants versus the growth rate (gray symbols), compared to wild-type *Escherichia coli* K-12 (white symbols, data from Fig 1). Death rate increases with growth rate with a smaller slope (*ΔrpoS*: $0.57 \pm 0.13$ h) compared to wild type (WT: $0.88 \pm 0.13$ h). Black and gray lines show linear weighted least square fits to log-transformed data, which take uncertainty in data points into account (see Methods). Data shown as mean $\pm$ SD. Two or three replicates per condition.

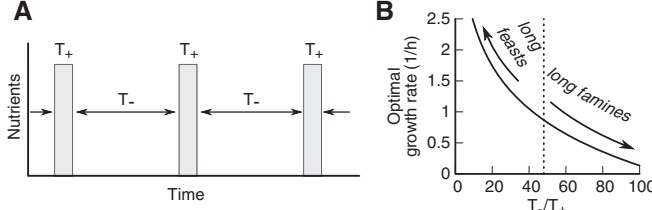

**Figure 5. Trade-off between growth and death in a fluctuating environment.**

A  A minimal ecological scenario of alternating periods of feast and famine is used to study overall fitness of different strategies. Periods of famine and feast can vary between scenarios. Based on the observed relation between growth and death (Figs 1 and 2), we assume that bacteria cannot independently choose growth and death rate, but are forced to trade-off growth against death according to the exponential relation $\gamma$ ($\mu$) = 0.21 day$^{-1}$ exp (1.0 h$\mu$).

B  The optimal growth rate, i.e., the one that maximizes fitness over a cycle of famine and feast, depends on the ratio of feast and famine periods. For long periods of feast interrupted by short periods of famine, the optimal strategy is to grow fast. For long periods of starvation intermitted by short periods of growth, the optimal strategy is long survival and slow growth. For the scenario of panels (A, B), $T_-/T_+ = 48$, the optimum is $\mu = 0.86$ h$^{-1}$ and $\gamma = 0.5$ day$^{-1}$, see dashed line.

the initial density $N(0)$, to reach a density $N(0)$ exp ($\mu T_+$). They will then die exponentially, at a rate that depends on the growth rate during feast, $\gamma(\mu)$. After a full cycle of famine and feast, the density of viable cells has changed from $N(0)$ to $N(0)$ exp ($\mu T_+ - \gamma(\mu)T_-$). Together, growth and death are therefore associated with the fitness

$$f(T_+, T_-, \mu) = \log(N(T_+ + T_-)/N(0)) = \mu T_+ - \gamma(\mu)T_-. \quad (1)$$

The growth rate $\mu*$ that maximizes fitness depends on the relative duration of famine and feast and can be calculated by taking the derivative of equation (1) with respect to the growth rate, $0 = \mathrm{d}f$ $(T_+, T_-, \mu)/\mathrm{d}\mu = T_+ - 0.21$ day$^{-1}$·1.0 h exp (1.0 h $\mu*$) $T_-$. Solving for $\mu*$, we find $\mu* = 1.0$ h$^{-1}$ ln ($114 \cdot T_+/T_-$), which is indicated as a dashed line in Fig 5B. For instance, given a $T_+ = 3$ h growth period and a $T_- = 3$ days starvation, maximal fitness is achieved with a growth rate $\mu = 0.86$ h$^{-1}$ and the associated death rate of $\gamma = 0.5$ day$^{-1}$. Even for longer growth periods, the optimal growth rate remains well below the maximal growth rate of *E. coli* in rich medium (~ 2.2 h$^{-1}$), suggesting a high selective pressure to reduce growth rate in anticipation of starvation.

## Global remodeling of proteome upon entry into starvation permits only partial adaptation

A salient question arising from the trade-off between growth and death in Fig 5B is to what extent bacteria can avoid the trade-off altogether, by adapting their proteome composition in the final phase of growth. In principle, *E. coli* could quickly grow to high cell density and then prepare for starvation by gradually slowing growth, exploiting its global regulation that adjusts proteome composition as a function of growth rate (Scott *et al*, 2010). However, global remodeling of the proteome is a slow process, since protein degradation does not contribute significantly, such that proteome composition adapts by dilution of the inherited proteome

with newly synthesized proteins (Erickson *et al*, 2017). It is therefore essential to clarify whether global remodeling of the proteome during the final phase of growth is sufficient to significantly alter the death rate during starvation.

To address this question, we describe the dynamics of proteome remodeling using the flux-controlled regulation (FCR) model established by Erickson *et al* (2017). This model uses metabolite fluxes to set regulatory functions and allows predictive modeling of growth and proteome composition without free parameters. We add to the FCR model our assumption that maintenance rate (and thus death rate) depends on the proteome composition (Fig 6A). Exploiting the observation that for most proteins, the proteome fraction $\phi_X$ depends linearly on the steady-state growth rate $\mu$ during exponential growth, $\phi_X = \phi_{X,0} + \mu\phi'_X$ (Hui *et al*, 2015), we define an instantaneous adaptation state $(\phi_X(t) - \phi_{X,0})/\phi'_X$ for each proteome sector. The value of this state variable corresponds to the steady-state growth rate $\mu$ that would yield a proteome sector of the current size $\phi_X$. Because of the global regulation of the proteome composition, most proteins adapt on the same time-scale (Erickson *et al*, 2017), and hence, a single state variable can capture the adaptation state of the proteome. The adaptation state must then also determine the maintenance rate $\beta$, and thus, the death rate via $\gamma = \beta/\alpha$, in a way consistent with our growth–death relation $\gamma(\mu) = 0.21 \, \text{day}^{-1} \exp(1.0 \, \text{h} \, \mu)$ established above. This yields an expression for the death rate as a function of the proteome adaptation state at the end of the growth phase,

$$\gamma(t) = 0.21 \, \text{day}^{-1} \exp\left(1.0 \, \text{h}(\phi_X - \phi_{X,0})/\phi'_X\right). \qquad (2)$$

Together with the FCR model, which describes the dynamics of $\phi_X(t)$ during growth, we thus obtain a complete description of the adaptation dynamics.

Using this mathematical model, we first investigate how much bacteria can adapt when the external nutrient concentration drops below the nutrient uptake affinity, see Fig 6B. The decrease in the nutrient concentration at the end of growth will slow down nutrient uptake and lead to a gradual entry to starvation. We model the uptake of a single nutrient to be dependent on the concentration of an external nutrient according to a Michaelis–Menten type function. To explore the range of possible behaviors, we compute the adaptation dynamics for three different Michaelis constants $K_M$ (10 μM, 100 μM, and 1 mM), which are in the range of typical uptake affinities ($\sim$ 5–200 μM). Lower affinities lead to a more gradual decrease in growth rate compared to higher affinities (Fig 6B, left). During this decrease in growth rate, the proteome can partially adapt (Fig 6B, center), and as a result, death rates are slightly smaller (Fig 6B, right). The inability of the bacteria to fully adapt arises because they grow by only a small amount during the slow growth period (small part of total protein per cell is synthesized after the external nutrient concentration is below $K_M$). For example, for an affinity of 1 mM and an initial nutrient concentration of 5 mM, cells only grow 1/5 doubling when the nutrient concentration is below the uptake affinity. This is insufficient to dilute the inherited proteome composition, and as a result, bacteria can only adapt slightly during the depletion of a single nutrient.

Compared to a single nutrient, a mixture of nutrients running out sequentially could allow better adaptation, because the growth rate can decrease more gradually. Entry of stationary phase and starvation by depletion of a mixture of nutrients is indeed also a typical scenario in studies of bacterial adaptation to starvation (Finkel, 2006; Hengge, 2011; Gefen *et al*, 2014) and leads to a substantial change in the proteome composition (Schmidt *et al*, 2016). To explore the effect of nutrient mixtures, we model a varying number of nutrients that gradually decrease the growth rate as they are sequentially depleted (Fig 6C, left). During this sequential depletion, bacteria can adapt their proteome better (Fig 6C, center) and reach noticeably lower death rates (Fig 6C, right). The reason for the improved adaptation compared to a single nutrient is that bacteria in complex environments produce a more substantial amount of biomass after the first nutrients have run out. This biomass is produced at slow growth and leads to a more thoroughly remodeled proteome.

An example for the sequential depletion of nutrients is the fermentation–respiration growth cycle, where microorganisms first ferment a carbon substrate and excrete a waste product such as acetate or ethanol, followed by respiration of the waste product and finally starvation. Because typically cells grow slower during respiration (Basan *et al*, 2017), this growth cycle leads to bi-phasic growth, similar to Fig 6C (green). In a recent paper, Li *et al* (2019) showed a trade-off for *Saccharomyces cerevisiae* during such a fermentation–respiration growth cycle between growth rate during respiration (the 2nd phase) and survival, but not between growth rate during fermentation (1st phase) and survival, indicating that microorganisms indeed can lose their proteomic "memory" during entry to stationary phase if they are allowed to adapt.

# Discussion

In this work, we reported a quantitative relation between the death rate of *E. coli* in carbon starvation and its growth rate prior to starvation. Environments that support slow growth lead to longer survival in starvation. After considering that the prior growth rate also affects cell size during starvation, we found that the effect on the death rate is primarily due to a change in the maintenance rate. This maintenance cost for a cell to preserve its viability during starvation increases exponentially with $\mu$, leading to the observed exponential increase in death rate $\gamma$ with the pre-starvation growth rate $\mu$. We explored the implications of the growth–death relation using a mathematical model, which predicts a strong selective pressure for *E. coli* to limit its growth rate. To some degree, cells can circumvent the trade-off between growth and death rate by adapting their proteome toward the end of growth phases, prior to entering starvation. However, our model illustrates that realistic scenarios for the adaptation process allow for only partial adaptation, and that growth on complex mixtures of nutrients leads to significantly better adaptation than growth on a single carbon source.

### Maintenance rate modulation as a survival strategy in energy-limited environments

A large proportion of bacteria on Earth live in energy-limited environments, such as the deep biosphere, where bacteria grow very slowly, with doubling times estimated between 1 and 3,000 years

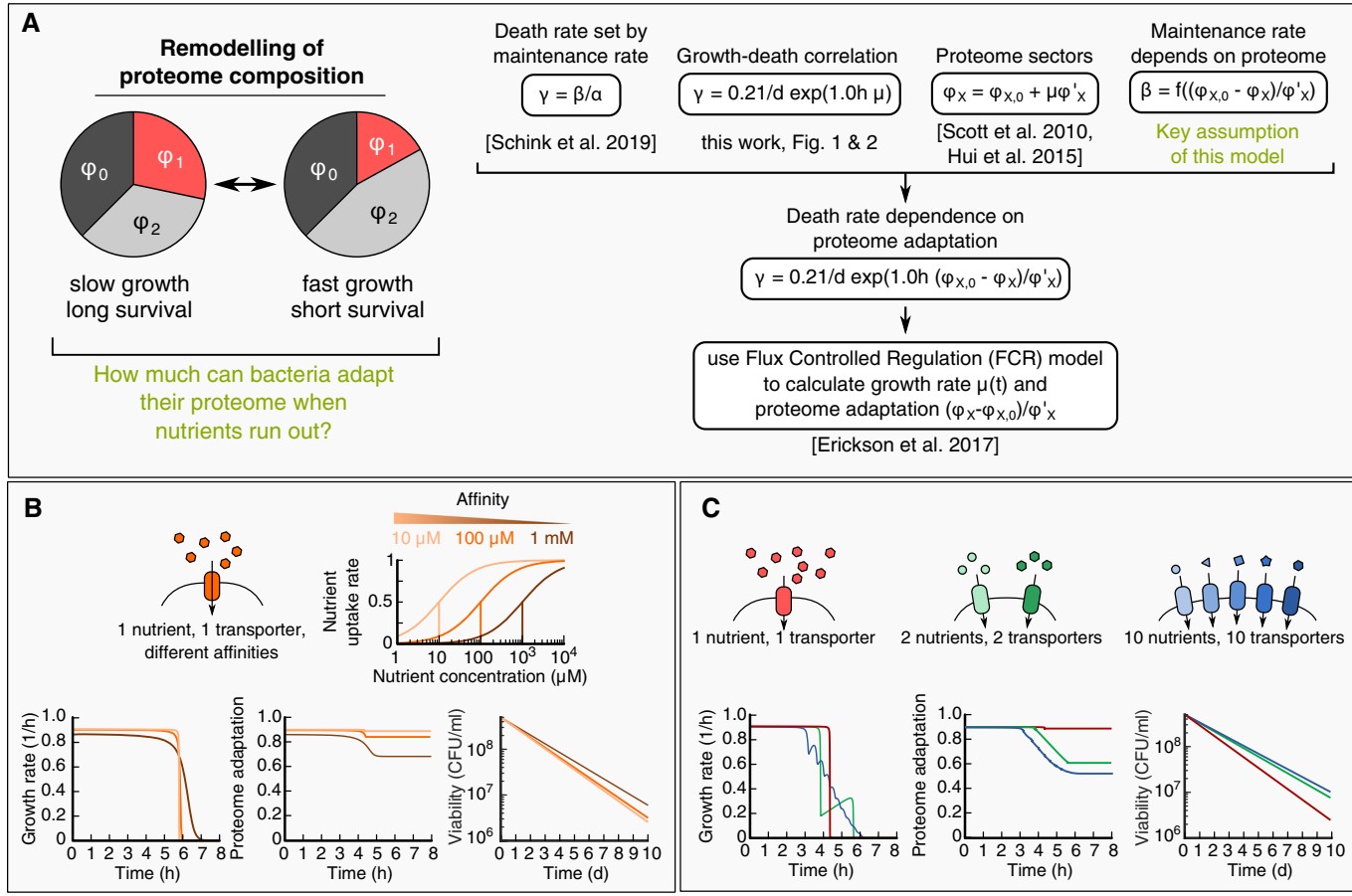

**Figure 6. Adaptability during entry to stationary phase.**

A  Mathematical modeling of death rate changes as bacteria adapt during entry to stationary phase, see Methods for details. The key assumption is that changes in maintenance rate $\beta$ are due to changes in the proteome. To understand adaption, we thus need to understand how much bacteria can remodel their proteome when nutrients run out (A – left). We combine our key assumption with the following quantitative laws: (i) Death rate is set by maintenance rate; (ii) Proteome sectors depend linearly on growth rate (Hui *et al*, 2015), and (iii) Adapt at the same time-scale during nutrient shifts (Erickson *et al*, 2017). Combined, these relations allow us to compute how death rate depends on the proteome adaptation (A – right). We use the flux-controlled regulation (FCR) model (Erickson *et al*), to calculate the relative adaptation $(\phi_x - \phi_{x,o})/\phi'_x$ in various scenarios.

B  Adaptation at the end of growth. When the nutrient concentration decreases, the nutrient uptake rate slows down with a Michaelis–Menten type kinetics (top). Depending on the uptake affinity, bacteria will experience either a sharp or smooth decrease in growth rate (bottom left, bright orange: 10 µM, dark orange: 1 mM). During this slow down, the proteome adapts (bottom center), with lower affinities (dark orange) showing a slight improvement of proteome adaptation, which results in a slightly lower death rate (bottom right).

C  Adaptation in complex media. A medium supplemented with multiple nutrients (one – red, two – green, ten – blue) leads to a step-wise decrease in growth rate during nutrient depletion (bottom left). Due to the extended periods of slow growth after exhaustion of primary nutrients, bacteria can adapt their proteome substantially (bottom center) and decrease their death rate (bottom right).

(Parkes *et al*, 1990; D'Hondt *et al*, 2002; Lomstein *et al*, 2012), or become dormant (Lennon and Jones, 2011). Their apparent ability to reduce the maintenance requirement to a bare minimum to survive these extreme energy-limited environments is not understood. An interesting observation by van Bodegom (van Bodegom, 2007) showed that maintenance rate (measured during growth) correlates with growth rate over a wide range of growth rates across different bacterial species, suggesting that different species have adapted to different levels of energy limitation. Our finding that *E. coli* has the ability to adapt its maintenance requirement then suggests that studying carbon starvation in single bacterial species under variable prior growth conditions is a promising approach to study the mechanisms of maintenance rate modulation. We hope

that our approach will be transferable to slow-growing bacterial species, in order to establish a quantitative basis for characterizing how vegetative bacteria survive environments with extreme energy limitation.

## Resource allocation during slow growth

In environments that support only slow growth, bacteria remodel their proteome and increase the expression of "starvation" proteins. Several of these "starvation inducible" proteins were described, from those driven by global regulators like CRP-cAMP, Lrp, and ppGpp to the general stress response regulated by RpoS (Hengge, 2011). In particular, RpoS is involved in improving

nutrient scavenging abilities (Franchini et al, 2015; Houser et al, 2015; Schmidt et al, 2016), improving cellular protection against potential harm such as heat shock, osmotic shock, oxidative damage, and acid stress (Hengge-Aronis, 1993; Hengge, 2011) and improved energy storage of glycogen or carbon residues (Hengge-Aronis & Fischer, 1992; Phaiboun et al, 2015). The growth–death relationship found in this work, however, appears not to hinge on RpoS, as slower growth of a knock-out of rpoS still leads to slower death.

In accordance with this result, RpoS was previously found to decrease cell maintenance rate by only 15% (Schink et al, 2019), much less than the 2.5-fold modulation of maintenance rate by growth rate observed in this work (Fig 3D). This does not mean that RpoS and the general stress response are generally dispensable in starvation. RpoS is responsible for the increase in biomass recycling (Schink et al, 2019), which manifests itself in the increase in the death rate displayed by the rpoS knock-out strain (Fig 4). Taken together, our results suggest that the molecular adaptation of cells to starvation is multifaceted, with RpoS-mediated regulation supported by a major proteome remodeling that decreases the cellular maintenance rate.

### Trade-off between survival and growth may shape bacterial fitness

As the cell adjusts its physiology toward growth, it becomes less adapted for survival. A conceivable origin of this trade-off is the proteome allocation problem of bacteria (Scott et al, 2010). Synthesizing proteins that protect cells and increase their survival chances comes at the expense of synthesizing proteins needed for growth. The result is a trade-off between a fitness benefit in starvation and a fitness cost during growth. Because fitness is the average proliferation across cycles of growth and death, fitness costs in one environment can be compensated by fitness benefits in others. As a result, investments in anticipation of changing environments can become a favorable strategy. This occurs, for example, for the overproduction of ribosomes that allows E. coli to quickly adapt to improved nutrient conditions (Koch, 1971; Li et al, 2018; Mori et al, 2017), the preparation of E. coli to challenging environments (Ghoul and Mitri, 2016) or antibiotic persistence (Balaban et al, 2004). Our finding that environments supporting only slow growth lead to longer survival thus suggests that E. coli is increasing its investment into survival by modulating the maintenance rate when growth is decreasing, i.e., when cells anticipate an approaching starvation. This trade-off between maintenance rate and the biosynthesis capacity may shed light on the long-standing questions why E. coli has not maximized its growth rate during evolution (Lenski et al, 1991; Basan et al, 2017; Towbin et al, 2017).

# Methods

### Contact for reagent and resource sharing

Further information and requests for resources and reagents should be directed to and will be fulfilled by the lead contact, Ulrich Gerland (gerland@tum.de).

### Experimental model and subject details

All strains used in this study are derived from wild-type E. coli K-12 strain NCM3722 (Soupene et al, 2003). The rpoS knock-out was transferred from JW5437-1 (Baba et al, 2006) to NCM3722 via P1 transduction to yield strain NQ1191. Glycerol kinase mutant GlpK22 (Pettigrew et al, 1996) was transferred via P1 transduction to yield strain NQ898.

### Culture medium

The culture medium used in this study is based on $N^-C^-$ minimal medium (Csonka et al, 1994), containing $K_2SO_4$ (1 g), $K_2HPO_4 \cdot 3H_2O$ (17.7 g), $KH_2PO_4$ (4.7 g), $MgSO_4 \cdot 7H_2O$ (0.1 g), and NaCl (2.5 g) per liter. The medium was supplemented with 20 mM $NH_4Cl$, as nitrogen source, and 5 mM glycerol, as the sole carbon source. One millimolar IPTG was added to media when necessary to fully induce the native lac operon. All chemicals were purchased from Carl Roth, Karlsruhe, Germany.

### Growth protocol

Before each experiment, cells were taken from −80°C glycerol stock and streaked out on an LB Agar plate. For growth in batch cultures, the same protocol described in Schink et al, 2019 was applied. However, due to acetate excretion in the medium in the presence of carbon sources different from glycerol and at growth rates higher than 0.7 $h^{-1}$ (e.g., growth of NQ898) (Basan et al, 2015), before entry into starvation, cells were centrifuged (3,000 RCF for 3 min), washed, and re-supplemented with medium free of carbon to avoid survival on waste products such as acetate. When no leftover nutrients are available, this washing step does not alter the physiology (Schink et al, 2019).

For the chemostat experiments, growth was carried out in three steps: seed culture and pre-culture in batch mode and continuous culture in chemostat. The first two steps were performed at 37°C in a water bath shaker at 250 rpm (WSB-30, Witeg, Wertheim, Germany) with water bath preservative (Akasolv, Akadia, Mannheim, Germany). The seed culture was prepared with fresh LB medium and inoculated with a single colony from the LB Agar plate. The pre-culture was performed in medium identical to the continuous culture, inoculated with a small amount of seed culture previously washed by centrifugation. The size of the inoculum was chosen such that the pre-culture grown overnight was still growing exponentially in the morning of the experiment. The seed culture was performed in 20 mm × 150 mm glass test tubes (Fisher Scientific, Hampton, NH, USA) with disposable, polypropylene Kim-Kap closures (Kimble Chase, Vineland, NJ, USA). The pre-culture had a volume of 200 ml and was performed in 500-ml baffled Erlenmeyer flasks (Carl Roth) and Kim-Kap closures. Once cells in the pre-culture have performed at least ten doublings and the optical density of the culture had reached the value $OD_{600} \approx 2$, 100 ml of the pre-culture was inoculated in 1 l of culture medium (see above) in the chemostat. Cells grew in such medium for ≈ 2 h in batch mode, reaching $OD_{600} \approx 0.5$ at the time of glycerol depletion. Then, continuous culture mode was applied and cells were grown in a constant volume of 1 l at 37°C with air-pressured spilling out the effluent at this volume, while the flow rate was controlled by the pump speed of the

incoming feed. The bioreactor used was a Infors HT, Labfors 5 with 2.0-l glass vessel, controlled by its intrinsic Control-Software. Air flow at 2 vvm and stirrer speed of 500 rpm were applied to keep the dissolved oxygen concentration with the relative $pO_2$ greater than 90% for all experiments, to avoid limiting in oxygen. pH was kept at $7.0 \pm 0.2$ by a pH probe and automatic addition of a solution of 2% $H_3PO_4$. Optical cell density was kept constant between 0.35 and 0.55. Different dilution rates were established by changing the pump speed, and the cultivations were performed from low to high growth rates (0.1, 0.2, 0.3, 0.4, 0.5, 0.7 $h^{-1}$). For each growth rate, steady-state growth was reached within few minutes after setting the growth rate. However, to obtain a complete turn-over of the cells in the culture, for each growth rate chosen, six generations were performed before extracting the sample for starvation. It has been reported that glucose-limited chemostat cultivations strongly select for loss or attenuation of RpoS function in *E. coli* so that mutations occur in *rpoS* after 30 generations (Notley & Ferenci, 1996). To avoid this possibility in our glycerol case, we did not run an experiment for more than 30 generations.

**Starvation protocol**

For each growth rate in batch culture and chemostat, cells were centrifuged (3,000 RCF for 3 min), washed, and re-suspended in minimal medium free of carbon in 20 mm × 150 mm glass test tubes (Fisher Scientific, Hampton, NH, USA) with disposable, polypropylene Kim-Kap closures (Kimble Chase, Vineland, NJ, USA). The tubes were placed in the water bath shaker at 250 rpm as the seed and pre-cultures and viability was measured by plating on LB Agar and counting the CFU after an incubation period of 12 h at 37 degrees. For each growth rate, three tubes of starved cultures were monitored in order to have three experimental repeats of cell behavior during starvation.

For each growth rate in the chemostat, after six generations, samples of the culture were extracted from the chemostat, using a SI Sample Syringe (20 ml, with 3-Way valve and check valve, C-Flex inlet) to avoid contaminations and to flush the line prior to sampling. In this way, the line being sampled did not have residual fluid left in it from an earlier sample. At the extraction time, a sample of the chemostat culture was immediately plated on LB Agar plates through dilution steps to check for contaminations/mutations in the chemostat. Then, cells were washed and starved as described above.

For growth rates 0.3 and 0.7 $h^{-1}$, two independent chemostat and starvation runs were performed. Samples were diluted in fresh $N^-C^-$ minimal medium without carbon substrate and spread on LB Agar using Rattler Plating Beads (Zymo Research, Irvine, CA, USA). LB Agar was supplemented with 25 µg $ml^{-1}$ of 2,3,5-triphenyltetrazolium chloride to stain colonies and increase contrast for automated colony counting (Scan 1200, Interscience, Saint-Nom-la-Bret_che, France) of 100–200 colonies per petri dish (92 × 16 mm, Sarstedt, Numbrecht, Germany). Staining or automation of counting had no significant effect on viability measurements or accuracy, compared to un-stained, manually counted samples (< 1% systematic error).

**Quantification of maintenance rate and yield**

Maintenance rate and yield were measured as described in Schink *et al* (2019). Measurements were performed for starved cultures of

wild-type *E. coli* K-12 NCM3722 previously grown in the chemostat at growth rates of 0.1, 0.3, 0.5, and 0.7 $h^{-1}$, respectively, and for the starved culture of NQ898 previously grown in batch mode at growth rate of 0.9 $h^{-1}$.

In order to quantify changes in the yield α, the growth yield during starvation was measured both for WT and Δ*rpoS* strains: At different times during starvation (from 0 to 7 days), 3–15 ml of the control culture was extracted, UV-sterilized, and mixed in a 99% to 1% ratio with untreated, starved culture. Viability of the growth curve $N_G$ was then measured every 12–24 h by plate counting. The recycling yield, α, was calculated as the ratio of the absolute growth yield, max $(N_G) - \min(N_G)$, where $\min(N_G)$ is the inoculation viability and max $(N_G)$ the maximal viability reached at the end of growth, to the viability in the control starved culture at the extraction point before killing (Fig 2E and F).

Uncertainty in recycling yield is estimated by assuming that "absolute yield" and "viability at extraction" can both only be measured up to 10%, due to errors in plate counting. Using the assumption that plating uncertainties in both quantities are uncorrelated, we propagate them to individual measurements of the recycling yield, i.e., individual data points in Fig 2E. Uncertainties in individual recycling yield measurements are propagated to the recycling yield shown in Fig 2F using the weighted least square fit described in the "Linear regression analysis" part of this Method section.

In order to quantify the maintenance β during starvation, at different times during starvation, samples of at least 3 ml were extracted from an exponentially decaying culture and a small amount of glycerol was added in each of them. The amount of glycerol was chosen in such a way that cells could not grow substantially using the supplied carbon substrate. In particular, at each time-point, in different samples, 10, 20, 30, and 40 µM of glycerol were added to cultures of ~ $10^8$ CFU $ml^{-1}$. After glycerol addition, cell viability was measured at least every 24 h by plate counting both in the control culture $N(t)$ and in the samples with glycerol added $N_{gly}(t)$. The lag time was then calculated as $T = \ln (\langle N_{gly}/N \rangle)/\gamma$, where the brackets denote an average over all time points after the initial lag, when all cultures are in exponential decay. Using the theoretical expression of $T$, derived in Schink *et al* (2019), for each day, β was extracted from the inverse of the slope of the linear fit of the experimental lag times plotted versus the amount of glycerol added per cell (Fig 2I and J). Uncertainties in maintenance rate are calculated by assuming that the underlying lag times can only be measured with 10% accuracy due to plating errors. Uncertainties in lag times are propagated to maintenance rate measurements.

**Live/dead stain**

Established commercial BacLight®LIVE/DEAD (Thermo Fisher Scientific Inc., Waltham, Massachusetts, USA) staining was used when cells were microscopically imaged, according to manufacturing specifications.

**Measurements of cell size**

To measure cell size, samples from batch or continuous cultures were extracted, stained with BacLight®LIVE/DEAD (Thermo Fisher

Scientific Inc., Waltham, Massachusetts, USA), placed on a cover slide, and imaged with phase-contrast microscopy, using a Nikon Ti microscope with a Plan Apo 100× oil objective (numerical aperture of 1.45 and a refractive index of 1.515). The used camera was an Andor Zyla VSC-02357, with a binning of 1 × 1, a readout rate of 200 MHz, and an exposure time of 200 ms. Conversion gain was set on 1/3 Dual gain, and the spurious noise filter was activated. The calibration from length units to pixel was defined as 0.07 μm px$^{-1}$. Measurements were performed with an activated perfect focus system, taking 10 × 10 image frames and moving the sample with a PriorScan III drive stage after each acquisition step. Cell areas were then manually determined using the Nikon software. For each growth rate, at least 300 cells were analyzed to determine averaged length and width. Cell volume was computed as $V = \pi(w/2)^2 (l - w) + 4/3\, \pi\, (w/2)^3$ where $w$ and $l$ are width and length of each cell, whose shape was considered as a cylinder [with base ray equal to $w/2$ and height equal to $(l - w)$] with two semi-spheres at the ends (with ray equal to $w/2$).

## Statistical analysis

Growth rates $\mu$ in the chemostat are reported with an error of 5%, estimated from pump accuracy. Growth rates $\mu$ in batch cultures and death rates $\gamma$ are the averages of at least three experimental repeats, reported with one standard deviation. Values of cell lengths, widths, and volume are the averages of 200 microscopy measurements. Recycling yield $\alpha$ and maintenance rate $\beta$ are the slopes of the fits (see "Linear regression analysis" in Methods) reported with standard error. Recycling yield $\alpha$ and maintenance rate $\beta$ normalized per cell volume are reported with error propagation, where the instrument error for the volume measurements was estimated as 5%.

## Linear regression analysis

To account for the uncertainties of the data points in linear regression analyses, we used weighted least squares fitting of linear regressions to log-transformed data (Press *et al*, 1986); i.e., we fitted a linear regression $y(x) = a + bx$ using weights $w_i = 1/(\sigma_{a,i}^2 + b\sigma_{b,i}^2)$ that depend on the standard deviation of the experimental data $\sigma_{a,i}$ and $\sigma_{b,i}$ of data points $i$ in $x$ and $y$ direction so that the resulting $\chi^2 = \sum_{i=1}^{N} w_i(y_i - a - bx_i)^2$ is minimized. Standard errors are calculated as the interval where $\chi^2$ takes a value that is 1 greater than at its minimum, $\Delta\chi^2 = 1$. The goodness-of-fit $Q$ is calculated as the probability that a value of $\chi^2$ as poor as found in the fitting procedure did not occur by chance.

## Modeling nutrient depletion

In order to describe the proteome remodeling of *E. coli* when nutrients run out, we use the FCR model described in Erickson *et al* (2017). The FCR model uses qualitative knowledge of the mechanistic regulation of catabolic and ribosomal proteins by cAMP and ppGpp, respectively, to construct regulation functions that yield predictive descriptions of growth and proteome composition during nutrient shifts. It was shown to accurately describe nutrient downshifts, including the bacterium's response to the gradual depletion of nutrients toward the end of growth. Here, we model carbon uptake

flux as the product of catabolic protein abundance $\phi_{cat}$ times an uptake rate $k$. This uptake rate depends on the external nutrient concentration with a Michaelis–Menten type function, $k(c) = k_{max}c/((c + K_M))$, where the nutrient affinity $K_M$ is typically on order of 5–200 μM (Erickson *et al*, 2017). The total amount of carbon influx is the sum of all catabolic proteins $M_{cat,j}$ times their respective uptake rate $k_j (c_j)$, $J(t) = \Sigma_J k_j (c_j) M_{cat,j} (t)$. In Fig 6C, we chose the uptake rates $k_j$ of different nutrients, such that growth rate on the remaining $m$ to $N$ nutrients decreases linearly, $\lambda_{m \to N} = \lambda_{max} \cdot (N - m - 1)/N$. Uptake rates $k_j$ are then calculated as

$$k_m = \frac{\lambda_{m \to N}}{1 - \lambda_{(m \to N)}/\lambda_C} - \sum_{j=m+1}^{N} k_j,$$

where $\lambda_C = 1.17$ h$^{-1}$ is the intercept of the C-line (You *et al*, 2013). The total carbon uptake determines growth of the biomass $M(t) = \int_{-\infty}^{t} J(t')dt'$. Growth rate $\mu(t) = J(t)/M(t)$ is plotted in Fig 6. In order to get the dynamics of $J(t)$, we calculate the change in abundance $M_{cat,j}$ from the FCR model. First, we calculate the translation activity as $\sigma = J/R$, where $R$ is the total ribosome abundance. We then use the translational activity $\sigma$ to determine the regulatory functions defined in Erickson *et al* (2017), $\chi_R (\sigma) = \phi_{Rb,0}/(1 - \sigma/\gamma)$ and $\chi_{Cat}(\sigma) = 1 - \sigma\hat{\chi}_{Rb}(\sigma)\lambda_C^{-1}$, where $\phi_{Rb,0}$ is the ribosome abundance interpolated to zero growth rate, $\gamma$ is the maximal translation rate, $\lambda_C = 1.17$ h$^{-1}$ is the intercept of the C-line (You *et al*, 2013). Next, we use the regulatory function to determine the synthesis of ribosomes $M_{Rb} = \chi_R (\sigma)J(t)$ and catabolic proteins $M_{Cat,j} = h_j \chi_{Cat} (\sigma)J(t)$. We solve the FCR model by integrating it numerically.

Proteome sectors typically depend linearly on growth rate, $\phi_X = \phi_X, 0 + \mu\phi_X'$, regardless of whether they increase or decrease with growth rate. This is not only true for ribosomes or catabolic proteins, but also true for the majority of other proteins (Hui *et al*, 2015). During growth shifts, the proteome composition adapts dynamically. This adaptation is controlled globally, with proteome sectors showing almost identical adaptation dynamics (Erickson *et al*, 2017). This is because of the proteome constraint; i.e., the sum of all proteome sectors has to equal one. This proteome constraint leads to the proteome sectors that increase or decrease to have matching dynamics, with only the sign and magnitude differing. As a result, we can define a relative adaptation of a proteome sectors $\phi_X$ (t), as $(\phi_X(t) - \phi_{X,0})/\phi_X'$. In exponential growth, the value of this "proteome adaptation" corresponds to the steady-state growth rate. After dynamic adaptation, the "proteome adaptation" equals the steady-state growth rate that would yield the same proteome composition. Thus, if *E. coli* initially grew at 0.9 h$^{-1}$ and the proteome adaptation changes to $(\phi_X - \phi_{X,0})/\phi_X' = 0.6$h$^{-1}$ after all nutrients have run out, this means that proteome has changed to the composition typically found at a growth rate of 0.6 h$^{-1}$.

If we assume that death rate is set by the proteome composition, either by proteome sectors increasing or decreasing with growth rate, then we can use the proteome adaptation to set death rate, by substituting steady-state growth rate $\mu$ with $(\phi_X - \phi_{X,0})/\phi_X'$, where we take the final $\phi_X$ after nutrient depletion. Using the exponential fit from Fig 2, we get $\gamma = 0.21$h$^{-1} \exp((\phi_X - \phi_{X,0})/\phi_X' 1.0$h$)$, which was used to generate the survival curves in Fig 6.

**Expanded View** for this article is available online.

## Acknowledgements

We thank Terence Hwa for providing strains NCM3722, NQ898, and NQ1191. We thank Constantin Ammar for his contributions in the early stage of this work. We thank Dirk Weuster-Botz for providing the bioreactor and its experimental setup in the Research Center for Industrial Biotechnology of the Technical University of Munich. We thank Dominik Maslak for assistance in the experiments performed with the bioreactor. This work was partially supported by the German Research Foundation (DFG) via the priority program SPP1617 and the Transregio 174 "Spatiotemporal dynamics of bacterial cells" through U.G. E.B. was partially supported by a DFG fellowship through the Graduate School of Quantitative Biosciences Munich (QBM). S.J.S is supported by a Long-Term Fellowship (ALTF 782-2017) from the European Molecular Biology Organization (EMBO).

## Author contributions

EB, SJS, and UG conceived the study and designed the experiments. EB performed the experiments with contributions from SJS EB, and SJS analyzed the experiments. SJS and UG developed the model. EB, SJS, and UG wrote the paper.

## Conflict of interest

The authors declare that they have no conflict of interest.

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
