## [Review Process File · Molecular Systems Biology]

Slower growth of *E. coli* leads to longer survival in carbon starvation

Elena Biselli, Severin Schink, and Ulrich Gerland

DOI: [10.15252/msb.20209478](https://doi.org/10.15252/msb.20209478)

Corresponding author(s): Ulrich Gerland (gerland@tum.de)

Review Timeline:

Initial Submission Date:	15th Nov 19
Editorial Decision:	19th Nov 19
Authors' Appeal:	22nd Nov 19
Editorial Decision:	3rd Dec 19
Second Submission Date:	27th Jan 20
Editorial Decision:	5th Mar 20
Revision Received:	8th Apr 20
Editorial Decision:	21st Apr 20
Revision Received:	28th Apr 20
Accepted:	29th Apr 20

Editor: Maria Polychronidou

Transaction Report:

Thank you for submitting your manuscript "Slower growth of E. coli leads to longer survival in starvation due to a decrease of the maintenance rate" to Molecular Systems Biology.

We have now considered your manuscript and I regret to inform you that we have decided to not send it out for peer review.

In this study, you examine how the growth conditions before carbon starvation affect the death rate of E. coli cells upon starvation. You observe that the death rate depends on the growth rate but not on the carbon source and you then examine the contribution of the maintenance rate and recycling yield, which you have previously found to determine the death rate (Schink et al, 2019). We appreciate that you report that this dependence is mainly explained by changes in the maintenance rate and we acknowledge the analysis of RpoS mutant E. coli, which indicates that the general stress response pathway is only partially responsible for the growth-death correlation, while other determinants remain to be investigated. However, we feel that the mechanistic insights into the observed dependencies remains rather limited while the broader implications of the presented findings for understanding bacterial fitness remain to be further demonstrated. Overall, we are not convinced that the study provides the degree of mechanistic insight and the level of conclusiveness that would be required for publication in Molecular Systems Biology.

That said, your work is a good candidate for Life Science Alliance (<http://www.life-science-alliance.org/>) our broad scope Open Access journal published in partnership between EMBO Press, Rockefeller University Press, and Cold Spring Harbor Laboratory Press. The editors of Life Science Alliance would be pleased to send your manuscript for peer review. Please use the following link to transfer your manuscript (no reformatting is required):

Link Not Available

I apologize for not being able to bring better news regarding the publication of your study at Molecular Systems Biology and I hope that you will view the possibility of a transfer to Life Science Alliance favorably.

Authors' Appeal**22nd Nov 19**

Thank you for evaluating our manuscript and for the explanation supplied with your decision email.

As just discussed over the phone, I am writing to you to emphasize one important point of our work that we may not have sufficiently stressed in our abstract and cover letter: Our experimental observation that the death rate of E. coli depends on its growth rate prior to starvation is much more than a mere correlation - instead, we identified a quantitative law that the death rate depends exponentially on the prior growth rate.

We then tracked down the physiological root of this law to the exponential dependence of the maintenance rate on the growth rate prior to starvation. Regarding molecular mechanism, we showed that there is not a single cause for the overall physiological effect by analyzing the RpoS mutant. Instead, the law appears to emerge from a global adaptation of the cell, akin to the "growth laws" that were previously identified for balanced growth.

Thank you for your message regarding our decision on your manuscript MSB-19-9365. I apologize for the delay in getting back to you, which was due to the delay in obtaining comments from our Editorial Advisory Board. I have now read your manuscript once again and I have considered the points raised in your appeal letter and during our phone call. I have also consulted with two members of our Editorial Advisory Board and I regret to say that the outcome was not positive. As you will see below, we felt that the study seems somewhat preliminary in its current form, but we would be open to considering an extended version, should that be of interest for you.

In this study, you examine how the growth conditions before starvation affect the survival of *E. coli* cells upon starvation. You observe that the death rate depends exponentially on the prior growth rate but not on the carbon source. Since you had previously discovered that the death rate is determined by the maintenance rate and recycling yield (Schink et al, 2019), you examine the contribution of these two components, and report that the dependence is mainly explained by changes in the maintenance rate. Finally, you perform analyses in RpoS mutant *E. coli*, which indicate that the general stress response pathway is only partially responsible for the growth-death rate dependence and that the underlying mechanism seems to be more complex. We think that the observed quantitative dependence of the death rate on the pre-starvation growth rate seems interesting. However, we still feel that in absence of either some level of mechanistic understanding of the observed dependencies, or perhaps more importantly, of a better understanding of the implications of this quantitative relationship for bacterial adaptation, the study remains somewhat preliminary.

Please note that we have also consulted two members of our Editorial Advisory Board. Both of them appreciated that the relation between the growth rate prior to starvation and the death rate upon starvation could be a potentially important finding. However, both of them mentioned that the study seems somewhat borderline for consideration at MSB. The two advisors had a slightly different take on why the study does not seem like a good fit in its current form. Specifically, one of them felt that the main finding (i.e. the quantitative relationship of growth rate and death rate) does not seem 'systems-level' enough for the journal, while the second advisor mentioned that either some level of mechanistic understanding or a more concrete demonstration of the implications of the study for understanding bacterial adaptation in changing environments and why bacteria in nature are not optimized for fast growth, would be required to make the study suitable for Molecular Systems Biology. As such, both advisors remained rather skeptical about sending the study out for review in its current form.

Taken together and based on the recommendation we received from two Editorial Advisory Board members I see no choice but to confirm our negative decision. However, since we and the two advisors thought that the main finding of the study is in principle interesting, we would be open to considering a new submission, including some follow up analyses in either of the directions delineated above. Specifically, such analyses could involve either some level of mechanistic insight or a demonstration of the implications for bacterial adaptation (either outside of a lab environment to understand the evolutionary implications in nature or in a lab evolution context). If you are interested in a potential resubmission and you would like to share your revision plans with me, I would be happy to discuss further.

I apologize for not being able to bring better news on this occasion and I do hope that you will consider Molecular Systems Biology for future submissions.

5th Mar 2020

Manuscript Number: MSB-20-9478

Title: Slower growth of *E. coli* leads to longer survival in starvation

Thank you again for submitting your work to Molecular Systems Biology. We have now heard back from the three referees who agreed to evaluate your study. Overall, the reviewers think that the presented findings seem interesting. They raise however a series of concerns, which we would ask you to address in a revision.

As you will see below, reviewer #1 thinks that the lack of a mechanistic understanding of the reported scaling law represents a limitation. This issue was also brought up in our previous editorial decision on the initial version of the work. However, as during our interactions after this initial submission we discussed that providing concrete mechanistic insights would be too involved at this stage and concluded that extending the study by including analyses of the implications of the growth-death relationship would seem a better option, and since the other two reviewers do not seem to be concerned about the lack of mechanism, we think that experiments dissecting the mechanism underlying the scaling law are not mandatory for the acceptance of the study. We would of course not be opposed to the inclusion of such data e.g. if you in the meantime happen to have it at hand. A short discussion on potential mechanisms, making it clear that it is speculative at this point, can be included.

The rest of the issues raised by the reviewers are rather clear and I think there is no need to repeat the points listed below. Please let me know in case you would like to discuss any of the issues raised by the reviewers.

On a more editorial level, we would ask you to address the following:

- Please provide a .doc version of the manuscript text (including legends for main figures, EV figures and tables) and individual production quality files for the main figures and EV figures.
- We have replaced Supplementary Information by the Expanded View (EV format). In this case, all additional figures and tables can be provided as EV Figures. EV Figures should be cited as 'Figure EV1, Figure EV2' etc... in the text and their respective legends should be included in the main text after the legends of regular figures. EV Tables should be provided as individual files, each containing the legend/description of the respective EV Table. For detailed instructions regarding expanded view please refer to our Author Guidelines: . In case several additional EV Figures are generated during revision (total number > 7), we would ask you to provide them in an Appendix PDF. Appendix figures should be labeled and called out as: "Appendix Figure S1, Appendix Figure S2... Appendix Table S1..." etc. Each legend should be below the corresponding Figure/Table in the Appendix. Please include a Table of Contents in the beginning of the Appendix.
- Please provide a "standfirst text" summarizing the study in one or two sentences (approximately 250 characters), three to four "bullet points" highlighting the main findings and a "synopsis image" (550px width and max 400px height, jpeg format) to highlight the paper on our homepage.

- All Materials and Methods need to be described in the main text. We would encourage you to use 'Structured Methods', our new Materials and Methods format. According to this format, the Material and Methods section should include a Reagents and Tools Table (listing key reagents, experimental models, software and relevant equipment and including their sources and relevant identifiers) followed by a Methods and Protocols section in which we encourage the authors to describe their methods using a step-by-step protocol format with bullet points, to facilitate the adoption of the methodologies across labs. More information on how to adhere to this format as well as downloadable templates (.doc or .xls) for the Reagents and Tools Table can be found in our author guidelines: . An example of a Method paper with Structured Methods can be found here: .

- Please include a Data availability section describing how the data and model generated in this study have been made available. This section needs to be formatted according to the example below:

The datasets and computer code produced in this study are available in the following databases:

- Chip-Seq data: Gene Expression Omnibus GSE46748

(<https://www.ncbi.nlm.nih.gov/geo/query/acc.cgi?acc=GSE46748>)

- [data type]: [full name of the resource] [accession number/identifier] ([doi or URL or identifiers.org/DATABASE:ACCESSION])

- When you resubmit your manuscript, please download our CHECKLIST

(<http://bit.ly/EMBOPressAuthorChecklist>) and include the completed form in your submission.

Please note that the Author Checklist will be published alongside the paper as part of the transparent process

(<https://www.embopress.org/page/journal/17444292/authorguide#transparentprocess>)

Please resubmit your revised manuscript online, with a covering letter listing amendments and responses to each point raised by the referees. Please resubmit the paper ****within one month**** and ideally as soon as possible. If we do not receive the revised manuscript within this time period, the file might be closed and any subsequent resubmission would be treated as a new manuscript. Please use the Manuscript Number (above) in all correspondence.

Click on the link below to submit your revised paper.

Link Not Available

If you do choose to resubmit, please click on the link below to submit the revision online before 4th Apr 2020.

Link Not Available

IMPORTANT: When you send your revision, we will require the following items:

1. the manuscript text in LaTeX, RTF or MS Word format
2. a letter with a detailed description of the changes made in response to the referees. Please specify clearly the exact places in the text (pages and paragraphs) where each change has been made in response to each specific comment given
3. three to four 'bullet points' highlighting the main findings of your study
4. a short 'blurb' text summarizing in two sentences the study (max. 250 characters)
5. a 'thumbnail image' (550px width and max 400px height, Illustrator, PowerPoint or jpeg format), which can be used as 'visual title' for the synopsis section of your paper.
6. Please include an author contributions statement after the Acknowledgements section (see <https://www.embopress.org/page/journal/17444292/authorguide#manuscriptpreparation>)
7. Please complete the CHECKLIST available at (<http://bit.ly/EMBOPressAuthorChecklist>). Please note that the Author Checklist will be published alongside the paper as part of the transparent process (<https://www.embopress.org/page/journal/17444292/authorguide#transparentprocess>).
8. Please note that corresponding authors are required to supply an ORCID ID for their name upon submission of a revised manuscript (EMBO Press signed a joint statement to encourage ORCID adoption) (<https://www.embopress.org/page/journal/17444292/authorguide#editorialprocess>).

Currently, our records indicate that there is no ORCID associated with your account.

Please click the link below to provide an ORCID:

Link Not Available

The system will prompt you to fill in your funding and payment information. This will allow Wiley to send you a quote for the article processing charge (APC) in case of acceptance. This quote takes into account any reduction or fee waivers that you may be eligible for. Authors do not need to pay any fees before their manuscript is accepted and transferred to the publisher.

***** PLEASE NOTE ***** As part of the EMBO Press transparent editorial process initiative (see our Editorial at <http://dx.doi.org/10.1038/msb.2010.72> , Molecular Systems Biology will publish online a Review Process File to accompany accepted manuscripts. When preparing your letter of response, please be aware that in the event of acceptance, your cover letter/point-by-point document will be included as part of this File, which will be available to the scientific community. More information about this initiative is available in our Instructions to Authors. If you have any questions about this initiative, please contact the editorial office (msb@embo.org).

Reviewer #1:

In their previous paper (Schick et al 2019, Cell Systems), the authors have established that the E. coli death rate can be described as ratio between maintenance rate and recycling yield. Now, the authors

- report that slower growth prior to starvation leads to a lower maintenance rate and lower death rates during starvation,
- find that RpoS can only partly explain the effect (on maintenance),
- report two explanatory models that put the findings in context.

This work is technically sound, and its description is clear. The question is whether the work represents sufficient advance to warrant publication in MSB.

The work presents a scaling law (i.e. the slower cells grow the more they scale down their maintenance "expenses" and thus they can live longer during starvation). This is what one would have expected. I very much acknowledge the dilemma here: In the biosciences, we most often strive for the "unexpected" because then we perceived work as "novel" (i.e. a new mechanism, a new posttranslational modification ...). In cases, where things make a lot of sense (i.e. where insights are consistent with researchers' intuitions), then we often consider work less novel, although in fact the ACTUAL piece of evidence has not been reported before. Anyway, let's consider the reported "scaling law" a novel finding.

Yet, for a publication in MSB I would expect some more insights in a paper. The presented models are explanatory and make a lot of sense. But again they don't provide much further new insight. The experiment with the rpoS deletion is nice, but it is only the very first step towards understanding of what is mechanistically responsible for the observed scaling law. I feel that for making this a suitable contribution for MSB, it would be necessary to add more mechanistic insights (beyond the already presented experiment with the RpoS deletion).

Questions:

- Usually maintenance rate is expressed in terms of ATP used per time and per biomass. If the authors convert their fmol glycerol/time/CFU into this unit: Are the numbers in the same order of magnitude? Are they consistent?
- The authors determine survival with CFU assays. Could it be that the actual survival rate is actually higher and only the re-growth capability on plate is decreased (i.e. that cells got sort of dormant and cannot regrow on plates)? I think it would be a good to also test viability with alternative means, e.g. live/dead stains.
- If I am not mistaken, then slower growing cells accumulate more storage compounds. Could this be a contributor for the observed correlation?
- The concentration of ppGpp has also been shown to correlate with growth rate (Lazzarini et al. 1971 J Biol Chem, 246, 4381-4385). Could this be one of the contributing factors?

Reviewer #2:

Understanding the general rules governing microbial growth and death is a central challenge in making progress in elucidating the evolutionary and ecological dynamics of microbial populations. This was a very interesting and clearly presented study of the effect of microbial growth rate on survival probability and resilience in starvation conditions. The authors found that faster-growing cells die exponentially faster when starved. At first this appears to be due to differences in both maintenance rate and recycling yield among fast and slow growers. But after adjusting for differences in cell volume among fast and slow growers, the exponential relation is found to be solely caused by differences in maintenance rate. The data were beautiful and clearly presented. In addition, their quantitative analysis is impressive and strengthens the paper's conclusions. I very

much liked this paper and feel that it could be published as is, but would nonetheless benefit from clarifying some questions described below.

Comments/questions:

1. The explanation of maintenance cost in lines 375-379 is nice, and would be better in the Introduction to provide better context. Additionally, I would appreciate more explanation in the Introduction of why only two factors (maintenance rate and recycling yield) were analyzed to understand changes in death rate.
2. More discussion of the chemostat assay would be nice; in particular, the authors should explicitly say that at steady state the division rate of the cells is equal to the dilution rate. However, the reason that this is true is because the resource concentration falls until those rates are equal, so in principle the authors could have just used different carbon source concentrations, right?
3. Line 374 mentions that doubling times may be up to 3000 years in the deep biosphere. Would it be possible for a low maintenance cost to achieve this much resilience? Would the authors models predict a continuous transition from low division rates to low death rates as a function of the carbon concentration becoming smaller?
4. The authors' model makes predictions about the proteome during stationary phase-is there any easy way to test this in experiments? I am not asking for new experiments, but it would be good to at least comment regarding what is known from the literature.
5. The starting densities in Fig. 1B vary, presumably due to different resource concentrations and yields. Does this difference matter to the interpretation of the data? They seem to be inversely related to growth rate in Fig. 1A, which is surprising to me.
6. In Fig. 2K: I think that y-axis should be gamma, not mu.
7. It is not obvious how to derive the expression for growth rate that maximizes fitness on line 262, and no supplemental section is referenced here.
8. In the results section describing the normalization of yield by cell size (line 211) I would have appreciated an additional sentence completing the chain of logic regarding what the authors think is going on. In particular, larger cells have more nutrients, but if the starving cells are also large then they should also need more nutrients. Is the difference because the cells become rapidly smaller during starvation?
9. In the Discussion section on trade-offs during different stages of growth death it reminded me of Li et. al. (2019, <https://www.nature.com/articles/s41559-019-0993-0>), who found Pareto fronts representing tradeoffs between different growth phases in yeast. They found tradeoffs between fermentation and respiration phases, and respiration and stationary phases, but not between stationary and fermentation phases. Since fermentation represents the most rapid growth, one might expect a tradeoff between stationary and fermentation phases, based on your results. Perhaps the lack of a tradeoff is due to the fact that yeast can sporulate to survive harsh conditions, or maybe your assay is better than theirs for evaluating this tradeoff. If the authors think that this is interesting / relevant they could include it, but not necessary.

Reviewer #3:

The authors extend their previous quantitative characterization of the death rate of E coli upon carbon starvation to a range of pre-starvation growth rates. They find a strong exponential dependence of the death rate on the pre-starvation growth rate (Fig 1C), which they factor in terms of recycling yield (scavenged energy from dead organisms; Fig 2F) and maintenance rate (energy

required to maintain viability; Fig 2J). The reasonable null-hypothesis that the variation in maintenance rate is regulated by RpoS is challenged by the observation of similar, though weaker, growth-rate dependence in the death rate of a *rpoS*- strain (Fig 4). This motivates a 'proteome-remodelling' mechanism to rationalize the coupling between pre-starvation growth rate and the death rate (Fig 6), and to predict the timescale for physiological adaptation to starvation.

I think this is an outstanding piece of work. The quantitative data, the experimental design and the internal self-consistency checks are of the highest standard. Furthermore, the mathematical modeling is fully-constrained by empirical measurements and yet provides deep insight into the origins of the growth rate-death rate coupling, and the unavoidable trade-off between growth and survival. The authors' rationalization of adaptation kinetics in terms of proteome remodelling could potentially clear away much of the historical fog surrounding transitions to stationary phase, allowing growth arrest to be studied with the same level of reproducibility that exponential growth enjoys.

Comments on content:

1. Throughout, the authors are careful to limit their claims to carbon starvation [that qualifier should perhaps be included in the title '...survival in *carbon* starvation due to...']. Historically, bacterial physiologists have avoided studying transitions out of exponential growth in part because Neidhardt and Magasanik showed [eg. Fig 1 of *Biochim. Biophys. Acta* v. 42 (1960), p. 99-116] that 'stationary phase' is not a well-defined concept, and that the physiology of the organism in stationary phase depends upon how growth arrest is achieved. It may be that the authors will find different empirical correlations among maintenance rate, recycling yield and pre-starvation growth rate if, for example, nitrogen or phosphate is exhausted in place of carbon. On the other hand, and this is the more exciting possibility, it may be that the variation observed by Neidhardt and Magasanik, and all subsequent work on transitions to stationary phase, is largely explained by variations in the transport K_m for the starvation cue and the subsequent duration of proteome remodelling.

2. Section starting on line 371: I agree with the authors that it is an important discovery that the maintenance rate can vary within the same species, and in a predictable fashion. In this section, however, the authors suggest that the copiotrophic bacterium *E. coli* is a useful model organism for inferring maintenance rate modulation in slow-growing oligotrophs [that is how I interpret the connection between lines 378-379 and lines 381-382]. I don't think there is any evidence presented here to support that claim. It would indeed be remarkable, and open up a whole new way of looking at maintenance rate, if it turned out that some (or all) oligotrophs exhibited similar empirical correlations between maintenance rate and pre-starvation growth rate, but there are too many untested jumps of inference to claim that the *E. coli* data can be generalized in this way. I think it is premature to suggest that chronic energy limitation is perceived by the organism in the same way as transient energy limitation. Similarly, lines 415-416 are overly-general: 'the cell' should be replaced with '*E. coli*'.

Comments on form:

1. Throughout, it would be useful to have error bars in the figures, particularly where the slope is being reported (Fig1C, Fig2BFJK, Fig3, Fig4). Likewise, it would be useful if the scaling relations on lines 360, 362, and 365 carried propagated uncertainties.

2. It is not clear from the Methods how the uncertainty in the slope is propagated. In the Methods (Line 741-743), it is reported that '[Parameters alpha and beta] are the values of the slope of [*exponential*] fits (see Methods) reported with 95% confidence bounds.' I find no other reference

in the Methods to the slope fits, so I assume the '(see Methods)' indicator is a typographical error.

Are the slopes fit using weighted-least squares on the logarithm of the data, with +/- two standard deviations? That is to say, are the uncertainties in the data points themselves propagated to the slope uncertainty, or is the quoted slope error simply the error associated with an exponential fit to error-free data?

3. The table data and the figure data don't appear to correspond in Fig1C-Table S1, and again in Fig 4-Table S4. The casamino acid figure data, in particular, appear to share the same death rate for glucose and glycerol in both wildtype and rpoS; that is not the case in the table data. Fig 4 shows 12, possibly 13, RpoS data points; Table S4 lists only 11. Furthermore, the error in the batch culture growth rate is reported throughout as +/- 0.01 /h. It seems unlikely that triplicate measurements over 24 conditions should return the same deviation. I assume this, too, is a typographical error.

Response to the Reviewers

Reviewer #1:

In their previous paper (Schick et al 2019, Cell Systems), the authors have established that the E. coli death rate can be described as ratio between maintenance rate and recycling yield. Now, the authors

- report that slower growth prior to starvation leads to a lower maintenance rate and lower death rates during starvation,
- find that RpoS can only partly explain the effect (on maintenance),
- report two explanatory models that put the findings in context.

This work is technically sound, and its description is clear. The question is whether the work represents sufficient advance to warrant publication in MSB.

The work presents a scaling law (i.e. the slower cells grow the more they scale down their maintenance "expenses" and thus they can live longer during starvation). This is what one would have expected. I very much acknowledge the dilemma here: In the biosciences, we most often strive for the "unexpected" because then we perceived work as "novel" (i.e. a new mechanism, a new posttranslational modification ...). In cases, where things make a lot of sense (i.e. where insights are consistent with researchers' intuitions), then we often consider work less novel, although in fact the ACTUAL piece of evidence has had not been reported before. Anyway, let's consider the reported "scaling law" a novel finding.

Yet, for a publication in MSB I would expect some more insights in a paper. The presented models are explanatory and make a lot of sense. But again they don't provide much further new insight. The experiment with the rpoS deletion is nice, but it is only the very first step towards understanding of what is mechanistically responsible for the observed scaling law. I feel that for making this a suitable contribution for MSB, it would be necessary to add more mechanistic insights (beyond the already presented experiment with the RpoS deletion).

We thank the Reviewer for these comments and the appreciation of the novelty of our work. We are also thankful for the thoughtful remarks about the dilemma regarding prioritization of "unexpected" results, and agree that more mechanistic insights would be desirable (as discussed further below). In an attempt to increase the Reviewer's appreciation of the package of results that we present in this manuscript, we'd like to illustrate and emphasize the added value offered by the quantitative nature of our 'scaling laws'.

We can think of quite a few examples of cases in the natural sciences, where progress was made in three stages that built on each other and occurred roughly as follows: First a discovery stage, then a stage of identifying quantitative phenomenological laws, and finally a stage of deep conceptual/mathematical understanding. A famous example from the early days of science is the discovery of planets, Kepler's phenomenological laws, and Newton's theory of gravitation. Or, in evolutionary genetics, the discovery of amino acid substitutions in proteins, the quantitative analysis of amino acid substitution rates in haemoglobin among different groups of animals, and then Kimura's theory of

neutral evolution, which could explain the quantitative behavior. (We'll discuss an example that is close to the subject of our manuscript in the next paragraph.) We believe it is fair to say that these three stages are equally important and equally valuable to the overall scientific progress. The results of the first stage are intrinsically novel and unexpected, while the subsequent stages two and three add considerable scientific value, but do not always add a novel phenomenon per se. In our eyes, our manuscript presents "stage two research", and as such should primarily be judged by asking "Does it take experimental observations to the level of quantitative phenomenological laws? If so, are these phenomenological laws likely to be useful for eventually obtaining a theoretical understanding with explanatory and predictive power."

We believe the answers to both questions are yes for our manuscript. Regarding the first one, we provide quantitative phenomenological laws stating that death rate and maintenance rate during starvation both depend exponentially on the prior growth rate (and we provide the corresponding exponential rate constants). For the second question, the answer is less clear, but we think the situation is comparable to the recent advances in the understanding of the regulation of bacterial growth, where the identification of quantitative phenomenological growth laws [Scott et al, Science (2010)] ultimately guided the researchers to a deep understanding of the regulatory mechanism and feedback scheme that coordinates the bacterial proteome with metabolism [You et al, Nature (2013)]. Similarly, we believe that our work is an important step towards a deep understanding of bacterial survival in carbon starvation, because it identifies the maintenance rate as a key property that *E. coli* can adapt. And the observation that this adaptation displays a simple exponential scaling with growth rate suggests that there is a central global feedback mechanism (akin to cyclic AMP signalling for growth) that performs this adaptation. This quantitative scaling law also serves as a test for candidate mechanisms to be discovered in the future: The disruption of the true core mechanism should eliminate the scaling behavior. By contrast, the RpoS deletion that we studied showed that while RpoS does affect the scaling behavior, it does not disrupt it and is therefore not at the core of the regulation scheme. Rather, it may be acting somewhere downstream from the core mechanism. In these ways, our results are enabling "stage three research".

Finally, we would like to emphasize that we also used mathematical modeling to clarify the *implications* of the identified growth-death trade-off for bacterial fitness and adaptation. Together, the presentation of the scaling law and its implications form a "package" that in our view should be compelling to a broad readership. As addressed below, the detailed comments of all three Reviewers also helped us to further improve this package.

Questions:

- Usually maintenance rate is expressed in terms of ATP used per time and per biomass. If the

authors convert their fmol glycerol/time/CFU into this unit: Are the numbers in the same order of magnitude? Are they consistent?

If we use a conversion of 15 ATP per molecule glycerol (Kaleta et al., 2013), a maintenance rate of 0.5 fmol glycerol /CFU*d converts into 7.5 fmol ATP/CFU*d = $5.2 * 10^4$ ATP/CFU*s. We can further convert CFU into mg dry weight by using 509 $\mu\text{g}/(\text{ml OD}_{600})$ from Ref. (Erickson et al 2017) and $1 \text{ OD}_{600} = 10^9 \text{ CFU}/\text{ml}$ to obtain $5.2 * 10^4 \text{ ATP}/(\text{CFU*s}) * 10^9 \text{ CFU}/(\text{ml OD}_{600}) * (\text{ml OD}_{600})/(509 \mu\text{g}) = 0.102 * 10^{12} \text{ ATP}/(\text{s } \mu\text{g}) = 368 * 10^{12} \text{ ATP}/(\text{h } \mu\text{g}) = 368 * 10^{18} \text{ ATP}/(\text{h g})$. Finally by converting ATP molecules into molar units using Avogadro's number $6.022 * 10^{23} /\text{mol}$ we obtain 0.61 mmole of ATP/((g dry weight) h). We have to take this number with caution, as we do not know what bacteria are using their nutrients for. ATP generation may only be partly responsible for the consumption of nutrients.

Comparing this number to the non-growth associated maintenance energy during exponential growth from Ref. (Varma et al 1994), 7.6 mmole of ATP/((g dry weight) h), or 8.4 mmole of ATP/((g dry weight) h) from Ref. (Feist et al 2007) our value is about 10 times lower. This is reasonable, given that the usual measurements of maintenance rate are performed during nutrient abundance, whereas we measure during starvation.

We agree that a comparison to maintenance measurements in exponential growth could be useful for the reader, so we included the following passage in the main text:

“We can compare our measurement of maintenance rate to literature values. Using a conversion of 15 ATP per molecule glycerol (Kaleta et al., 2013), a dry mass per OD of $509 \mu\text{g ml}^{-1} \text{ OD}_{600}^{-1}$ (Erickson et al, 2017) and $10^9 \text{ CFU ml}^{-1} \text{ OD}_{600}^{-1}$, we calculate that a maintenance rate of $0.5 \text{ fmmol (glycerol) d}^{-1} \text{ CFU}^{-1}$ corresponds to 0.61 mmole of ATP/((g dry weight) h). In comparison, estimates for non-growth associated maintenance energy during exponential growth for *E. coli* are about 10-fold higher at 7.6 (Varma et al 1994) and 8.4 mmole of ATP/((g dry weight) h) (Feist et al 2007). This shows that maintenance measurements during nutrient abundance are not readily convertible to starvation.”

- The authors determine survival with CFU assays. Could it be that the actual survival rate is actually higher and only the re-growth capability on plate is decreased (i.e. that cells got sort of dormant and cannot regrow on plates)? I think it would be a good to also test viability with alternative means, e.g. live/dead stains.

We did such a control in our previous work (Schink et al, Cell Systems, 2019, Figure S1), where we used the classic propidium iodide stain, which measures permeability as a “dead” stain. We found that the decay rate of an unstained “live” population and the decay rate of a population that formed colonies on LB agar plates agree with each other. After several days in starvation the vast majority of bacteria stained positive for

permeability, excluding the possibility that a majority of the population enters a dormant state.

We now refer the reader to our previous work in the beginning of the results section, to clarify that plate counting is a good measure of viability:

“[...] We then follow the survival kinetics by measuring bacterial viability via plate counting at different time points after carbon starvation (Fig. 1B). *Bacterial viability measurements by live/dead staining yields survival kinetics comparable to plate counting (Schink et al, 2019)*. The number of ‘colony forming units’ (CFU) per ml decreases exponentially for all cultures. [...]”

- If I am not mistaken, then slower growing cells accumulate more storage compounds. Could this be a contributor for the observed correlation?

We thank the Reviewer for this suggestion. We raised this point in the introduction where we cited Fung et al 2013, who measured glycogen storage in different growth phases. There are at least two ways in which carbon storage could influence survival. First, *E. coli* could use its stored carbon during the initial phase of starvation. This would be consistent with the observed lag in death (Schink et al, Cell Systems, 2019, Figure S1) during the first ~12 hours of starvation. After the storage has been used up, the population then starts dying exponentially at the usual rate. The second possibility is that individual bacteria do not use their own storage, but release it upon death as a public good. This would increase the amount of released nutrient, thus increase their recycling yield and decrease their death rate. Because we do not see a substantial change in recycling yield as a function of growth rate, we believe that no substantial nutrient storage is released during death. But since we have no measurement of the glycogen storage, we would like to refrain from making a statement in the manuscript.

- The concentration of ppGpp has also been shown to correlate with growth rate (Lazzarini et al. 1971 J Biol Chem, 246, 4381-4385). Could this be one of the contributing factors?

We also think that ppGpp is likely to be involved in the core regulatory mechanism responsible for the scaling law, since it is one of the master regulators in *E. coli* and, as pointed out by the Reviewer, it correlates with growth rate (and thus death rate). The issue is that such correlations, which are also observed for other regulators (cAMP, rpoS, etc.) and in fact large parts of the proteome (Hui et al, MSB, 2015), cannot separate between cause and consequence. We currently do not see a straightforward path to get to the core of the regulatory mechanism, but hope to find one in the future.

Reviewer #2:

Understanding the general rules governing microbial growth and death is a central challenge in making progress in elucidating the evolutionary and ecological dynamics of microbial populations. This was a very interesting and clearly presented study of the effect of microbial growth rate on survival probability and resilience in starvation conditions. The authors found that faster-growing cells die exponentially faster when starved. At first this appears to be due to differences in both maintenance rate and recycling yield among fast and slow growers. But after adjusting for differences in cell volume among fast and slow growers, the exponential relation is found to be solely caused by differences in maintenance rate. The data were beautiful and clearly presented. In addition, their quantitative analysis is impressive and strengthens the paper's conclusions. I very much liked this paper and feel that it could be published as is, but would nonetheless benefit from clarifying some questions described below.

We are grateful to the Reviewer for the appreciation of our work and for the valuable feedback below on how to improve our manuscript.

Comments/questions:

1. The explanation of maintenance cost in lines 375-379 is nice, and would be better in the Introduction to provide better context.

We agree with the Reviewer and shifted this aspect to the introduction.

Additionally, I would appreciate more explanation in the Introduction of why only two factors (maintenance rate and recycling yield) were analyzed to understand changes in death rate.

We followed the Reviewers' suggestion and now introduce our approach of measuring yield and maintenance rate in the introduction. This part now reads:

“During starvation, the resource which bacteria use for maintenance are nutrients recycled from dead bacteria. As a result, both changes in maintenance rate or recycling yield can alter the survival kinetics. To dissect the individual contributions of maintenance rate and recycling yield to death rate, we use the quantitative approach of Schink et al. (2019).”

2. More discussion of the chemostat assay would be nice; in particular, the authors should explicitly say that at steady state the division rate of the cells is equal to the dilution rate. However, the reason that this is true is because the resource concentration falls until those rates are equal, so in principle the authors could have just used different carbon source concentrations, right?

The Reviewer is right to point out that we did not fully describe the rationale here. The reason we could not have used different carbon concentrations is that due to the low K_m of glycerol ($\sim 5 \mu\text{M}$), we would have needed to use extremely low glycerol concentrations (down to $1 \mu\text{M}$), which yield only very low cell densities (OD 0.0001). Because at these low concentrations nutrients gradually run out (K_m comparable to glycerol concentration, a point discussed in Fig. 6B), we would have need to wash the culture at an OD below 0.0001, which is hard to do accurately. Switching to a nutrient with higher K_m (e.g. lactose or gluconate) might have been an option. However, such experiments would not guarantee the cells to be in a steady-state growth state. Running a chemostat in steady state allows us to grow *E. coli* at high densities at constant, low glycerol concentrations. This allowed us to maintain our established protocols of measuring maintenance rate and yield.

We included the following statement in the manuscript to explain our motivation of using a chemostat.:

“[...] In order for maintenance measurements of different cultures to be quantitatively comparable, this carbon source needs to be the same for all experiments. For this reason, we turn to a chemostat setup to change growth rate on the same carbon substrate. We choose glycerol as the limiting nutrient in the medium and let the bacterial culture grow in steady state at fixed dilution rate, such that dilution rate equals growth rate (see Methods). In this setup, glycerol concentration will be constant and low enough to reduce growth rate ($< 5 \mu\text{M}$), but bacterial density can be kept comparable to batch experiments ($5 \cdot 10^8 \text{ CFU/ml}$). We use the chemostat to vary growth rate from 0.1 h^{-1} to 0.7 h^{-1} .”

3. Line 374 mentions that doubling times may be up to 3000 years in the deep biosphere. Would it be possible for a low maintenance cost to achieve this much resilience?

We believe it is conceivable that bacteria can decrease their maintenance cost drastically. The extreme limit would be bacterial endospores, which seem to have zero maintenance cost, but aren't vegetative. If the maintenance rate of vegetative cells can be decreased to nearly zero, then these cells can use all remaining nutrients for growth. In this case a minute and/or intermittent carbon influx could be used to support very slow growth, e.g. on average one doubling every 3000 years.

Our main motivation to mention the deep biosphere was to make the reader aware that some organisms can survive in a vegetative state for millennia and that the fast death rates that we observe in the lab show only a certain fraction of the biologically attainable space of growth and death.

Would the authors models predict a continuous transition from low division rates to low death rates as a function of the carbon concentration becoming smaller?

Unfortunately, our models are not able to reliably predict the behavior in this regime. We treat growth independent from starvation. There is also the possibility of a mixed phase of coexisting growth and death in the transition region mentioned by the Reviewer.

4. The authors' model makes predictions about the proteome during stationary phase-is there any easy way to test this in experiments? I am not asking for new experiments, but it would be good to at least comment regarding what is known from the literature.

The ideal experiments would correlate changes in the proteome with survival rates. Substantial reorganization of the proteome, for example during entry to stationary phase, have been accurately measured in the literature.

We included the following statements highlighting the work done on proteome adaptation during the entry to stationary phase:

“Entry of stationary phase and starvation by depletion of a mixture of nutrients is indeed also a typical scenario in studies of bacterial adaptation to starvation (Gefen et al., 2014; Hengge, 2011; Finkel, 2006), and leads to substantial change in the proteome composition (Schmidt, 2015).”

5. The starting densities in Fig. 1B vary, presumably due to different resource concentrations and yields. Does this difference matter to the interpretation of the data? They seem to be inversely related to growth rate in Fig. 1A, which is surprising to me.

Growth was typically terminated at of about OD of 0.5 by washing the cultures and resuspending them into pre-warmed carbon free minimal medium. The change in density, which is inversely correlated to growth rate comes from a change in bacterial cell size. OD is a good measure for (dry) biomass, see e.g. Erickson et al Nature (2017) ED Fig. 1a, so we essentially starved cultures of the same biomass. Because faster growing bacteria are bigger, we see that the initial density of viable cells decreases for faster growing bacteria. Because there is no difference in death rates when *E. coli* is starved by washing, this difference does not matter for the interpretation of the data.

To make this point clearer, we included the following statement in the legend of Fig. 1: “Initial density for fast growth (lighter shades) is lower than for slow growth (darker shades), due larger cell sizes that lead to lower cell densities per OD₆₀₀.”

6. In Fig. 2K: I think that y-axis should be gamma, not mu.

We thank the Reviewer for pointing this out.

7. It is not obvious how to derive the expression for growth rate that maximizes fitness on line 262, and no supplemental section is referenced here.

We now derive the growth rate that maximizes fitness in the main text:

“The growth rate μ^* that maximizes fitness depends on the relative duration of famine and feast and can be calculated by taking the derivative of Eq. (1) with respect to growth rate,

$0 = df(T_+, T_-, \mu)/d\mu = T_+ - 0.21 \text{ d}^{-1} \cdot 1.0 \text{ h} \exp(1.0 \text{ h} \mu^*) T_-$. Solving for μ^* , we find $\mu^* = 1.0 \text{ h}^{-1} \ln(114 \cdot T_-/T_+)$, which is indicated as a dashed line in Fig. 5B.”

8. In the results section describing the normalization of yield by cell size (line 211) I would have appreciated an additional sentence completing the chain of logic regarding what the authors think is going on. In particular, larger cells have more nutrients, but if the starving cells are also large then they should also need more nutrients. Is the difference because the cells become rapidly smaller during starvation?

To clarify this, we added the following two sentences at the end of the section to which the Reviewer referred:

“This means that while bigger cells contain proportionally more nutrients, they use more than one would expect from their size for maintenance. This implies that there are factors involved beyond cell size that modulate the maintenance requirement of *E. coli*. ”

9. In the Discussion section on trade-offs during different stages of growth death it reminded me of Li et. al. (2019, <https://www.nature.com/articles/s41559-019-0993-0>), who found Pareto fronts representing tradeoffs between different growth phases in yeast. They found tradeoffs between fermentation and respiration phases, and respiration and stationary phases, but not between stationary and fermentation phases. Since fermentation represents the most rapid growth, one might expect a tradeoff between stationary and fermentation phases, based on your results. Perhaps the lack of a tradeoff is due to the fact that yeast can sporulate to survive harsh conditions, or maybe your assay is better than theirs for evaluating this tradeoff. If the authors think that this is interesting / relevant they could include it, but not necessary.

We thank the Reviewer for pointing out the relation to the work of Li et al (2019). According to our model, we would actually not expect a trade-off between fermentation and survival in stationary phase, if the slow-growing respiration phase in between is long enough to dilute the inherited fermentation-proteome. Our model therefore seems to be in accordance with the observation by Li et al. In the revised manuscript, we now relate to the results of Li et al at the end of the Results section, after presenting the results of Fig. 6C:

“An example for the sequential depletion of nutrients is the fermentation-respiration growth cycle, where microorganisms first ferment a carbon substrate and excrete a waste product such as acetate or ethanol, followed by respiration of the waste product and finally starvation. Because typically cells grow slower during respiration (Basan et al 2017), this growth cycle leads to bi-phasic growth,

similar to Fig. 6C (green). In a recent paper, Li et al (2019) showed a trade-off for *Saccharomyces cerevisiae* during such a fermentation-respiration growth cycle between growth rate during respiration (the 2nd phase) and survival, but not between growth rate during fermentation (1st phase) and survival, indicating that microorganisms indeed can lose their proteomic 'memory' during entry to stationary phase if they are allowed to adapt."

Reviewer #3:

The authors extend their previous quantitative characterization of the death rate of E coli upon carbon starvation to a range of pre-starvation growth rates. They find a strong exponential dependence of the death rate on the pre-starvation growth rate (Fig 1C), which they factor in terms of recycling yield (scavenged energy from dead organisms; Fig 2F) and maintenance rate (energy required to maintain viability; Fig 2J). The reasonable null-hypothesis that the variation in maintenance rate is regulated by RpoS is challenged by the observation of similar, though weaker, growth-rate dependence in the death rate of a rpoS- strain (Fig 4). This motivates a 'proteome-remodelling' mechanism to rationalize the coupling between pre-starvation growth rate and the death rate (Fig 6), and to predict the timescale for physiological adaptation to starvation.

I think this is an outstanding piece of work. The quantitative data, the experimental design and the internal self-consistency checks are of the highest standard. Furthermore, the mathematical modeling is fully-constrained by empirical measurements and yet provides deep insight into the origins of the growth rate-death rate coupling, and the unavoidable trade-off between growth and survival. The authors' rationalization of adaptation kinetics in terms of proteome remodelling could potentially clear away much of the historical fog surrounding transitions to stationary phase, allowing growth arrest to be studied with the same level of reproducibility that exponential growth enjoys.

We thank the Reviewer for this concise summary of our key results, the appreciation of our work, and the constructive feedback below.

Comments on content:

1. Throughout, the authors are careful to limit their claims to carbon starvation [that qualifier should perhaps be included in the title '...survival in *carbon* starvation due to...].

We changed our title to include 'carbon' starvation.

Historically, bacterial physiologists have avoided studying transitions out of exponential growth in part because Neidhardt and Magasanik showed [eg. Fig 1 of Biochim. Biophys. Acta v. 42 (1960), p. 99-116] that 'stationary phase' is not a well-defined concept, and that the physiology of the organism in stationary phase depends upon how growth arrest is achieved. It may be that the authors will find different empirical correlations among maintenance rate, recycling yield and pre-starvation growth rate if, for example, nitrogen or phosphate is exhausted in place of carbon. On the other hand, and this is the more exciting possibility, it may be that the variation observed by Neidhardt and Magasanik, and all subsequent work on transitions to stationary phase, is largely explained by variations in the transport K_m for the starvation cue and the subsequent duration of proteome remodelling.

We thank the Reviewer for these farsighted comments. The prospect of studying different types of starvation (= different ways of how “growth arrest is achieved”) is exciting. In this work we focused on carbon starvation, because *E. coli* needs the carbon substrate to generate energy. Absence of other growth limiting nutrients, e.g. limiting nitrogen or limiting arginine/guanine in an auxotrophic strain like in Neidhardt and Magasanik’s work would lead to growth arrest, while bacteria are presumably still able to generate energy.

If stationary phase is reached by depletion of carbon substrates, we believe that the transition phase in which carbon concentrations fall below the K_m could account for at least parts of the variation observed in the literature. We hope that experiments using ‘sharp’ transitions from growth to non-growth will help “clear the fog”, as the Reviewer put it.

After reading this comment of the Reviewer, we realized that the distinction between ‘carbon starvation’ (with the intention of limiting energy generation) and other types of starvation was not made very clearly in our manuscript. For this reason, we added the following statement in the introduction:

“ [...] Either by culturing *E. coli* in media with different carbon substrates, [...] or by varying the dilution rate in a carbon-limited chemostat. We then rapidly deprive each culture of its carbon substrate (*i.e. its energy source*) and measure its survival kinetics [...]

2. Section starting on line 371: I agree with the authors that it is an important discovery that the maintenance rate can vary within the same species, and in a predictable fashion. In this section, however, the authors suggest that the copiotrophic bacterium *E. coli* is a useful model organism for inferring maintenance rate modulation in slow-growing oligotrophs [that is how I interpret the connection between lines 378-379 and lines 381-382]. I don't think there is any evidence presented here to support that claim. It would indeed be remarkable, and open up a whole new way of looking at maintenance rate, if it turned out that some (or all) oligotrophs exhibited similar empirical correlations between maintenance rate and pre-starvation growth rate, but there are too many untested jumps of inference to claim that the *E. coli* data can be generalized in this way. I think it is premature to suggest that chronic energy limitation is perceived by the organism in the same way as transient energy limitation. Similarly, lines 415-416 are overly-general: 'the cell' should be replaced with '*E. coli*.'

We agree with the Reviewer that this part of the Discussion section was not sufficiently carefully worded. We did not mean to suggest that studying maintenance rate modulation in *E. coli* permits any immediate conclusions on how slow-growing oligotrophs deal with chronic energy limitation. Instead, this part was meant as an outlook, suggesting to transfer *our approach* (rather than our findings) to slow-growing species. In this context, we also relate to the data in Fig. 1 of (van Bodegom, 2007), which shows a striking cross-species correlation between growth rate and maintenance

rate (measured during growth) spanning both slow and fast-growing species, suggesting that there is a continuum of maintenance behaviors. We propose that varying growth conditions for a single species to vary its maintenance rate during starvation is a powerful approach to elucidate mechanisms of reducing maintenance requirements, because data are more easily comparable, as opposed to varying maintenance rates in different organisms. We adapted the wording of this part of the Discussion to reflect the speculative nature of our outlook:

“A large proportion of bacteria on Earth live in energy limited environments, such as the deep biosphere, where bacteria grow very slowly, with doubling times estimated between 1 and 3000 years (Parkes et al., 1990; Lomstein et al., 2012; D’Hondt et al., 2002), or become dormant (Lennon and Jones, 2011). Their apparent ability to reduce the maintenance requirement to a bare minimum to survive these extreme energy limited environments is not understood. An interesting observation by van Bodegom (van Bodegom, 2007) showed that maintenance rate (measured during growth) correlates with growth rate over a wide range of growth rates across different bacterial species, suggesting that different species have adapted to different levels of energy limitation. Our finding that *E. coli* has the ability to adapt its maintenance requirement then suggests that studying carbon starvation in single bacterial species under variable prior growth conditions is a promising approach to study the mechanisms of maintenance rate modulation. We hope that our approach will be transferable to slow growing bacterial species, in order to establish a quantitative basis for characterizing how vegetative bacteria survive environments with extreme energy limitation.”

Comments on form:

1. Throughout, it would be useful to have error bars in the figures, particularly where the slope is being reported (Fig1C, Fig2BFJK, Fig3, Fig4). Likewise, it would be useful if the scaling relations on lines 360, 362, and 365 carried propagated uncertainties.

As suggested, we now include error bars in Fig1C, Fig2BFJK, Fig3, and Fig4. Concerning the uncertainties, please see next comment.

2. It is not clear from the Methods how the uncertainty in the slope is propagated. In the Methods (Line 741-743), it is reported that '[Parameters alpha and beta] are the values of the slope of [*exponential*] fits (see Methods) reported with 95% confidence bounds.' I find no other reference in the Methods to the slope fits, so I assume the '(see Methods)' indicator is a typographical error. Are the slopes fit using weighted-least squares on the logarithm of the data, with +/- two standard deviations? That is to say, are the uncertainties in the data points themselves propagated to the slope uncertainty, or is the quoted slope error simply the error associated with an exponential fit to error-free data?

So far, we used least squares fitting on log-transformed data, which does not take the uncertainty of the data points into account. Given the Reviewer's comment, we changed our fitting procedure to weighted least squares fit, which we now explain in a new subsection of the Methods section.

Due to the concerns raised by the Reviewer, we also revisited how we calculate uncertainties of recycling yields and maintenance rates. This data is inherently noisy due to plating error, that's why in our work we needed to collect several measurements for the same condition (Fig. 2E and Fig. 2I), to be able to draw solid conclusions. In the revised version of the manuscript, we now assume that all parameters determined by plating, such as 'absolute yield', 'viability at extraction' and 'lag time' can only be determined with an uncertainty of 10%. We now propagate this uncertainty into the uncertainty of 'recycling yield' and 'maintenance rate' using the method described above. We note that due to the inclusion of the uncertainty of the data, the uncertainty of the reported exponential slopes has increased (as it should). We thank the Reviewer for this comment, which has improved the statistical rigor of our analysis.

3. The table data and the figure data don't appear to correspond in Fig1C-Table S1, and again in Fig 4-Table S4. The casamino acid figure data, in particular, appear to share the same death rate for glucose and glycerol in both wildtype and rpoS; that is not the case in the table data. Fig 4 shows 12, possibly 13, RpoS data points; Table S4 lists only 11. Furthermore, the error in the batch culture growth rate is reported throughout as +/- 0.01 /h. It seems unlikely that triplicate measurements over 24 conditions should return the same deviation. I assume this, too, is a typographical error.

We thank the Reviewer for catching these errors. Table S1 and S4 should have reported growth rates with one standard deviation, instead of a blanket +/- 0.01 /h. This is now fixed. Fig. 1C and Fig. 4 were created with a wrong table of data, which is now fixed.

21st Apr 2020

Manuscript Number: MSB-20-9478R

Title: Slower growth of E. coli leads to longer survival in carbon starvation

Thank you for sending us your revised manuscript. We think that the performed revisions satisfactorily address the issues raised by the reviewers. As such, I am glad to inform you that your manuscript is now suitable for publication, pending some minor editorial issues listed below.

We would ask you to address the following in a minor revision:

- Our data editors have noticed some unclear or missing information in the figure legends, please see the attached .doc file. Please make all requested text changes using the attached file and *keeping the "track changes" mode* so that we can easily access the edits made.

- Please provide individual files for the EV Tables.

- Please provide 5 keywords.

- Please format the reference list according to the MSB style i.e. listing the first 10 authors followed by et al.

- The provided "synopsis image" is rather complex, could you please send us an updated and simplified image? The image should provide a general idea of what the study shows, without too much detail and ideally with as little text as possible. The size of the synopsis image needs to be 550 px (width) x max 400 px (height).

- I have slightly edited the synopsis text (see attached file), could you please let me know if it is OK like this or if you would prefer to change anything?

- Due to the quantitative nature of the study we would encourage you to provide the Source Data for the Figure panels showing essential quantitative information. Source Data for main figures should be provided in .zip Folders labeled "Source data for Figure X". Please provide one .zip folder for each of the main figures. Source Data for Appendix Figures should all be provided in one single .zip folder labeled "Source Data for Appendix". Further information regarding Source Data can be found here: .

Please resubmit your revised manuscript online, with a covering letter listing amendments and responses to each point raised by the referees. Please resubmit the paper ****within one month**** and ideally as soon as possible. If we do not receive the revised manuscript within this time period, the file might be closed and any subsequent resubmission would be treated as a new manuscript. Please use the Manuscript Number (above) in all correspondence.

Click on the link below to submit your revised paper.

Link Not Available

If you do choose to resubmit, please click on the link below to submit the revision online before 21st May 2020.

Link Not Available

IMPORTANT: When you send your revision, we will require the following items:

1. the manuscript text in LaTeX, RTF or MS Word format
2. a letter with a detailed description of the changes made in response to the referees. Please specify clearly the exact places in the text (pages and paragraphs) where each change has been made in response to each specific comment given
3. three to four 'bullet points' highlighting the main findings of your study
4. a short 'blurb' text summarizing in two sentences the study (max. 250 characters)
5. a 'thumbnail image' (550px width and max 400px height, Illustrator, PowerPoint or jpeg format), which can be used as 'visual title' for the synopsis section of your paper.
6. Please include an author contributions statement after the Acknowledgements section (see <https://www.embopress.org/page/journal/17444292/authorguide#manuscriptpreparation>)
7. Please complete the CHECKLIST available at (<http://bit.ly/EMBOPressAuthorChecklist>). Please note that the Author Checklist will be published alongside the paper as part of the transparent process (<https://www.embopress.org/page/journal/17444292/authorguide#transparentprocess>).
8. Please note that corresponding authors are required to supply an ORCID ID for their name upon submission of a revised manuscript (EMBO Press signed a joint statement to encourage ORCID adoption) (<https://www.embopress.org/page/journal/17444292/authorguide#editorialprocess>).

Currently, our records indicate that the ORCID for your account is 0000-0002-0859-6422.

Link Not Available

The system will prompt you to fill in your funding and payment information. This will allow Wiley to send you a quote for the article processing charge (APC) in case of acceptance. This quote takes into account any reduction or fee waivers that you may be eligible for. Authors do not need to pay

any fees before their manuscript is accepted and transferred to the publisher.

*** PLEASE NOTE *** As part of the EMBO Press transparent editorial process initiative (see our Editorial at <http://dx.doi.org/10.1038/msb.2010.72> , Molecular Systems Biology will publish online a Review Process File to accompany accepted manuscripts. When preparing your letter of response, please be aware that in the event of acceptance, your cover letter/point-by-point document will be included as part of this File, which will be available to the scientific community. More information about this initiative is available in our Instructions to Authors. If you have any questions about this initiative, please contact the editorial office (msb@embo.org).

The authors performed the requested editorial changes.

29th Apr 2020

Manuscript number: MSB-20-9478RR

Title: Slower growth of E. coli leads to longer survival in carbon starvation

Thank you again for sending us your revised manuscript. We are now satisfied with the modifications made and I am pleased to inform you that your paper has been accepted for publication.

*** PLEASE NOTE *** As part of the EMBO Publications transparent editorial process initiative (see our Editorial at <http://dx.doi.org/10.103/msb.2010.72>), Molecular Systems Biology publishes online a Review Process File with each accepted manuscript. This file will be published in conjunction with your paper and will include the anonymous referee reports, your point-by-point response and all pertinent correspondence relating to the manuscript. If you do NOT want this File to be published, please inform the editorial office at msb@embo.org within 14 days upon receipt of the present letter.

LICENSE AND PAYMENT:

All articles published in Molecular Systems Biology are fully open access: immediately and freely available to read, download and share.

Molecular Systems Biology charges an article processing charge (APC) to cover the publication costs. You, as the corresponding author for this manuscript, should have already received a quote with the article processing fee separately.

Please let us know in case this quote has not been received.

Once your article is at Wiley for editorial production, you will receive an email from Wiley's Author Services system, which will ask you to log in and will present you with the publication license form for completion. Within the same system the publication fee can be paid by credit card, an invoice or pro forma can be requested.

Payment of the publication charge and the signed Open Access Agreement form must be received before the article can be published online.

Upon acceptance it is mandatory for you to return the completed payment form. Failure to send back the form may result in a delay in the publication of your paper.

Molecular Systems Biology articles are published under the Creative Commons licence CC BY, which facilitates the sharing of scientific information by reducing legal barriers, while mandating attribution of the source in accordance to standard scholarly practice.

Proofs will be forwarded to you within the next 2-3 weeks.

Thank you very much for submitting your work to Molecular Systems Biology.

Corresponding Author Name: Ulrich Gerland

Manuscript Number: MSB-2020-9478